# AgentsNet: Coordination and Collaborative Reasoning in Multi-Agent LLMs

## Abstract

Large-language models (LLMs) have demonstrated powerful problem-solving capabilities, in particular when organized in multi-agent systems. However, the advent of such systems also raises several questions on the ability of a complex network of agents to effectively self-organize and collaborate. While measuring performance on standard reasoning benchmarks indicates how well multi-agent systems can solve reasoning tasks, it is unclear whether these systems are able to leverage their topology effectively. Here, we propose AgentsNet, a new benchmark for multi-agent reasoning. By drawing inspiration from classical problems in distributed systems and graph theory, AgentsNet measures the ability of multi-agent systems to collaboratively form strategies for problem-solving, self-organization, and effective communication given a network topology. We evaluate a variety of baseline methods on AgentsNet including homogeneous networks of agents which first have to agree on basic protocols for organization and communication. We find that some frontier LLMs are already demonstrating strong performance for small networks but begin to fall off once the size of the network scales. While existing multi-agent benchmarks cover at most 2–5 agents, AgentsNet is practically unlimited in size and can scale with new generations of LLMs. As such, we also probe frontier models in a setup with up to 100 agents.

## 1 Introduction

Human societies thrive on collaboration, with language serving as the primary medium through which individuals coordinate and achieve collective goals. From small teams to large-scale organizations, effective communication enables structured decision-making, problem-solving, and the emergence of complex behaviors that surpass the capabilities of any single individual. This interplay between communication and coordination is mirrored in computing, where distributed systems rely on structured information exchange to tackle problems that exceed the capacity of any single processor. Just as psychology studies individual cognition while sociology examines emergent behaviors in groups, distributed systems research focuses on multi-agent coordination beyond what a single machine can accomplish (Lenzen & Wattenhofer, 2012).

Recently, distributed systems have been playing an increasingly important role in AI through the emergence of general-purpose multi-agent systems built on top of large language and vision models (LLMs). Agent-based frameworks such as generative agents (Park et al., 2023) have demonstrated the potential of solving complex problems with LLM-based agents. In particular, it has been shown that networks of LLM-based agents can outperform single agents (Hong et al., 2024; Qian et al., 2024a; Chen et al., 2024a; Qian et al., 2024b; Zhuge et al., 2024; Marro et al., 2024), mirroring aspects of human teamwork. For example, GPTSwarm (Zhuge et al., 2024) introduce a graph-based approach inspired by language-based societies of mind (Zhuge et al., 2023), demonstrating that organizing LLM-based agents in structured topologies enhances their performance on benchmarks like MMLU (Hendrycks et al., 2021), HumanEval (Chen et al., 2021), and GAIA (Mialon et al., 2024). MultiAgentBench (Zhu et al., 2025) aims to test collaboration, but is limited to a small number of agents and uses centralized shared memory. MACNET (Qian et al., 2024b) evaluates collaborative problem solving under DAG-structured communication with supervisory critic agents and global topological scheduling, which fundamentally differs from the fully decentralized, cycle-dependent setting studied in distributed computing. MAS-GPT (Ye et al., 2025), in turn, focuses on generating complete multi-agent systems in a single forward pass rather than evaluating coordination

among concurrently acting agents, and therefore does not address decentralized message passing or local-neighborhood decision making. Despite promising results from structured agent networks, existing benchmarks fall short in evaluating the core competencies of multi-agent systems: scalable coordination to a large number of nodes, decentralized communication, and collaborative reasoning. To address this gap, we introduce AGENTSNET, a multi-agent benchmark that measures these capabilities across diverse network structures and scales.

AGENTSNET assesses the agent's coordinative and collaborative capabilities through fundamental problems in distributed computing. Concretely, we identify five central problems from the distributed systems literature to construct corresponding coordination and collaboration tasks for multi-agent systems. Solving these tasks requires anything from local information aggregation to global coordination over multiple communication rounds. As a canonical example, whenever multi-agent systems are tasked with solving a certain problem, agents must necessarily be able to reach an agreement on the solution, a problem known in fault-tolerant distributed computing as consensus (Fischer et al., 1985). In another example, agents first agree on a single agent to take a leadership role, and then subsequently solve the task, guided and instructed by the elected leader. Selecting a single leader in a network is known as the leader election problem (Angluin, 1980). Fortunately, such problems are well-studied and theoretically grounded, providing an ideal testbed for the coordination and collaboration skills of multi-agent systems.

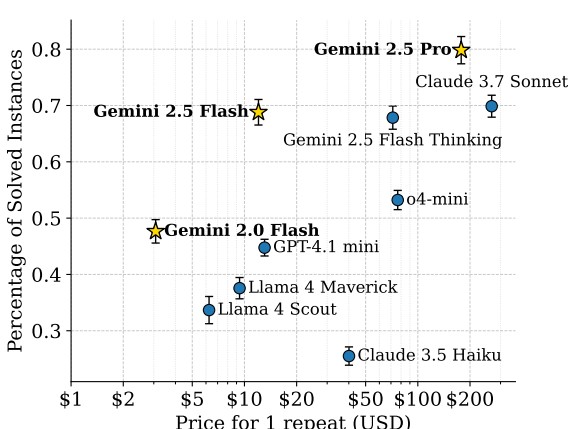

Figure 1: Mean AGENTSNET score of models versus API costs per repeat (May 15, 2025). Error bars indicate standard error of the mean. Gold stars denote Pareto-optimal models.

Various multi-agent benchmarks exist, but no benchmark explicitly assesses the ability of multi-agent systems for structured coordination and collaboration in a decentralized system, which should be seen as fundamental capabilities of effective distributed systems. As such, AGENTSNET complements the existing suite of multi-agent benchmarks of LLMs with a particular focus on grounding in distributed systems theory, network topology, and scalability to large agent networks. Concretely, we make the following contributions:

1. We build AGENTSNET from *graph coloring (resource allocation), minimal vertex cover (strategic positioning), maximal matching (bilateral negotiation), leader election (symmetry breaking and forming hierarchy)* and *consensus (global agreement)*: five fundamental distributed computing problems that evaluate the ability of multi-agent systems to test capabilities that are necessary for multi-agent systems.

2. We design a robust and scalable message-passing protocol for effective agent-to-agent communication and evaluate on a rich set of graph instances, sampled from various graph models such as small-world (Watts & Strogatz, 1998) or preferential attachment models (Barabási & Albert, 1999), which capture structural properties of real-world networks.

3. We evaluate a variety of agentic baselines on AGENTSNET, ranging from open-source LLMs such as Llama 4, to frontier models such as GPT, Gemini, and Claude, as well as the latest reasoning models, on the graphs of 4, 8, 16 nodes scaling the problem size to 100 agents which is well beyond existing agentic benchmarks.

4. We provide an in-depth qualitative analysis and highlight the challenges in coordinative and collaborative capabilities of LLMs to further improve multi-agent systems.

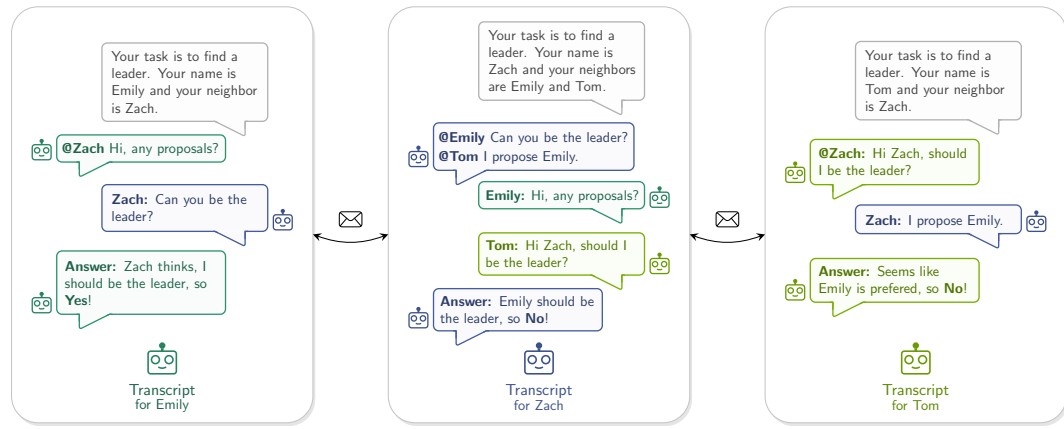

Figure 2: Example communication between three agents on a simplified topology. Agents Emily, Zach, and Tom each receive and send messages to their neighbors in multiple rounds of message-passing; see Section E for an in-depth qualitative analysis of transcripts.

## 2 RELATED WORK

Ensembling multiple agents to collaboratively negotiate solutions has emerged as an effective paradigm to improve LLM performance on complex tasks (Du et al., 2023; Xiong et al., 2023; Liang et al., 2024). This has been extended through work on different network topologies for more structured agent interaction. Some studies examine pre-determined graph structures (Hong et al., 2024; Qian et al., 2024a; Regan et al., 2024; Qian et al., 2024b) while others propose automatically adapting network topology (Liu et al., 2023; Chen et al., 2024a; Zhuge et al., 2024). Experiments show different topologies perform best for specific tasks (Chen et al., 2024a; Zhuge et al., 2024) and large-scale LLM agent networks exhibiting known social phenomena (Yang et al., 2024; Chuang et al., 2024). Parallel research examines LLMs' ability to reason with graph-structured data. Studies propose evaluation datasets (Fatemi et al., 2024; Wang et al., 2024; Zhang et al., 2024; Tang et al., 2025; Skianis et al., 2024) using single-agent setups where graphs are encoded as text. Fatemi et al. (2024) investigate graph encoding methods, Sanford et al. (2024) categorize graph reasoning problems by complexity, while Wang et al. (2024) and Skianis et al. (2024) explore effective prompting techniques. Our work bridges these research directions by studying multi-agent systems solving graph reasoning problems collaboratively. Our benchmark is complementary to recent agentic benchmarks (Liu et al., 2024; Yin et al., 2024; Agashe et al., 2024; Yao et al., 2024; Ni et al., 2025) but scales to a practically unlimited number of agents due to the generative problem creation protocol, with experiments involving up to 100 coordinating agents. Human studies on decentralized problem-solving in social networks show that network topology and size strongly influence coordination success (Kearns et al., 2006; Judd et al., 2010; Chiang et al., 2024). Section F provides an extended discussion of related studies.

## 3 TASKS, EVALUATION, AND GRAPH MODELS

To evaluate the ability of multi-agent systems to self-organize, coordinate, and communicate effectively, we design a benchmark consisting of fundamental problems from distributed computing. These problems span a range of complexities, from local tasks that require minimal coordination to global problems that necessitate multi-round communication. In what follows, we introduce the theoretical problems and describe how we map each problem to a corresponding agentic task. Afterwards, we introduce the graph distributions used within AGENTSNET.

### 3.1 BENCHMARKING TASKS

We evaluate multi-agent systems on a set of distributed computing problems that test their ability to aggregate information, self-organize, and coordinate. These tasks are selected for their foundational

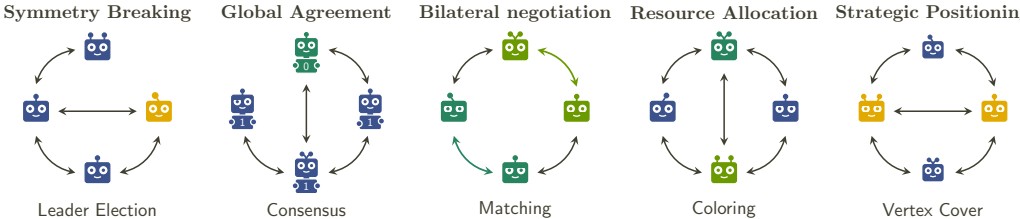

Figure 3: Overview of the tasks in AGENTSNET: In LEADERELECTION, the task is to select a single agent as the leader of the network. In CONSENSUS, the task is for all agents to agree on a specific value, for example 0 or 1. In MATCHING, the task is for pairs of agents to team up without conflicts. In COLORING, the task is for agents to select a group (indicated by a color), such that none of their neighbors are in the same group as them. In VERTEXCOVER, the task is to find a minimal group of coordinator agents such that each agent is a neighbor to at least one coordinator. Colors in the icons illustrate the roles or solution states relevant to each task (e.g., matched pairs, chosen colors, coordinator nodes) and differ across tasks accordingly.

nature in distributed computing and the core capabilities that they represent in multi-agent systems (like resource allocation for coloring). They span a diverse range of coordination requirements and communication complexities, from purely local information exchange to global decision-making; see Table 1 for an overview of the different theoretical problems selected for AGENTSNET.

**($\Delta + 1$)-Coloring: Resource Allocation.** Each node is assigned a color using at most $\Delta + 1$ colors, where $\Delta$ is the maximum node degree. This problem has a well-defined distributed complexity of $O(\log^* n)$ in bounded-degree graphs (Barenboim, 2016). This task is particularly useful for role assignment within multi-agent systems. For instance, agents can be designated to perform specific sub-tasks (e.g., web search, reasoning, coding, planning), with the constraint that directly connected agents are assigned distinct roles to avoid redundancy. Solving this task reflects the system's ability to efficiently distribute responsibilities across the network with minimal overlap in capabilities. The corresponding agentic task is to form groups, with a pre-defined number of groups, and where each group corresponds to a color. After message-passing, each agent chooses the group it wants to be in. The task is solved if the groups form a valid $\Delta + 1$-coloring. In AGENTSNET, we refer to this task as COLORING.

**Minimal Vertex Cover: Stategic Positioning.** A minimal vertex cover is a subset of nodes such that every edge in the graph has at least one endpoint in the subset, and removing any node from this subset would violate that property. This problem has a close relationship with the maximal independent set and is similarly fundamental in distributed computing, with known randomized solutions in $O(\log^* n)$ rounds Linial (1992). In agentic networks, a minimal vertex cover can represent a minimal set of monitor or gateway agents that maintain awareness of all interactions in the system. These agents could take on responsibilities such as relaying messages, auditing behavior, or bridging subgroups. The task tests the ability to identify a compact yet effective set of nodes with high influence or observability. The corresponding agentic task is to select a group of coordinators among the agents. After message-passing, each agent is asked whether it is a coordinator. The agents can respond with either *Yes* or *No*. The task is solved if coordinators form a minimal vertex cover. In AGENTSNET, we refer to this task as VERTEXCOVER.

**Maximal Matching: Bilateral Negotiation.** A maximal matching is a set of edges such that no two edges share a vertex, and no additional edges can be added without violating this property. This task captures the ability of agents to negotiate pairwise agreements without global knowledge, which is useful in scenarios where resource allocation or mutual exclusivity must be enforced (e.g., agent-to-agent task assignment). Randomized algorithms typically solve this problem in $O(\log^* n)$ rounds Peleg (2000). The corresponding agentic task is for the agents to form pairs. After message-passing, each agent is asked to name the neighbor it wants to pair up with. The agents can also respond with *None* if they cannot find a match (all neighbor agents are already paired up with other

agents). The task is solved if the paired agents form a maximal matching. In AGENTSNET, we refer to this task as MATCHING.

**Leader Election: Symmetry Breaking and Forming Hierarchy.** One node must be selected as the leader, while all others acknowledge that they are not. This classic coordination task is central to evaluating how well agents establish hierarchy and delegate global decision-making (Angluin, 1980). In multi-agent systems, leader election can be interpreted as selecting a central planner or controller agent responsible for strategy synthesis, while the remaining agents act as executors. Effective leader election demonstrates the system's capacity to break symmetry and converge on a single authority. In general graphs, the round complexity is $O(D)$, where $D$ is the network diameter Lynch (1996). The corresponding agentic task is to select a single leader among the agents. After message-passing, each agent is asked whether it is the leader. The agents can respond with either *Yes* or *No*. The task is solved if there exists exactly one leader. In AGENTSNET, we refer to this task as LEADERELECTION.

**Consensus: Global Agreement.** In the consensus problem, all agents must agree on a single value from the set $\{0, 1\}$. In our benchmark, we focus on the basic setting without any faulty or Byzantine agents. The goal is for all agents to coordinate and produce the same final answer after a number of communication rounds. A successful solution requires that every agent outputs the same value, either 0 or 1. This task tests the ability of multi-agent systems to converge to a global agreement through local message-passing alone. In synchronous networks, achieving consensus generally requires $\mathcal{O}(D)$ rounds Lynch (1996). The corresponding agentic task is to choose between a value 0 and 1. After message-passing, each agent is asked to announce its selected value. The task is solved if all agents announce the same value. In AGENTSNET, we refer to this task as CONSENSUS.

| Graph Problem | Round Complexity |
|---|---|
| $(\Delta + 1)$-Coloring | $\Omega(\log^*(n))$ |
| Minimal Vertex Cover | $\Omega(\log^*(n))$ |
| Leader election | $\Omega(D)$ |
| Maximal Matching | $\Omega(\log^*(n))$ |
| Consensus | $\Omega(D)$ |

Table 1: Overview of the theoretical problems from distributed computing that form the basis of AGENTSNET, together with (not necessarily tight) theoretical lower bounds for their round complexity in the randomized LOCAL (Linial, 1992) model.

Together, these tasks cover a broad spectrum of problems known in the distributed computing literature, which allows AGENTSNET to evaluate the reasoning, communication, and organizational capabilities of multi-agent systems.

## 3.2 NETWORK TOPOLOGIES

While classical distributed computing often studies problems on random graphs such as Erdős-Renyi networks (Erdos et al., 1960), these do not adequately capture the structural properties of real-world networks. Instead, we focus on three well-established graph models, namely the Watts-Strogatz graphs (Watts & Strogatz, 1998) (SMALLWORLD) exhibiting both short average path lengths and high clustering coefficients; preferential attachment graphs (Barabási & Albert, 1999) (SCALEFREE) containing hubs (high-degree nodes) and follow a power-law degree distribution; geometric graphs by constructing a Delaunay triangulation over randomly sampled 2D points, (DELAUNAY), maintaining a spatial relationship between nearby agents. We describe these graph models in more detail in Section D.

## 4 AGENT-TO-AGENT COMMUNICATION VIA MESSAGE-PASSING

To systematically study how agents exchange information and collaborate, we employ a communication model that draws inspiration from classical distributed computing, while adapting to the capabilities and constraints of modern LLM-based agents. Our setup is based on the LOCAL model (Linial, 1992) from distributed algorithms, in which the computation proceeds in synchronous rounds and each agent can exchange messages only with its immediate neighbors on the communication graph. Agents must base their decisions exclusively on local information aggregated over multiple rounds of interaction. This model captures fundamental aspects of decentralized reasoning,

where global strategies emerge from purely local exchanges without centralized control. Unlike nodes in deterministic systems, LLM-based agents exhibit stochastic behavior due to inherent randomness in their generation processes. This means that our model is most closely aligned with the randomized version of the LOCAL model. Given a communication network, each node, that is, each agent, is instantiated as an instruction-tuned LLM that interfaces with its neighbors through a structured chat history. Initially, we provide each agent with a *system prompt* detailing the task, for example, COLORING, the rules of message-passing, the names of its neighbors, and a notification that the agent must output a result in its *final response* after a fixed number of rounds of message-passing; see Section A for the full system prompt.

**Task Description.**  For each task, we provide a short description of the task, as well as which information we seek to extract in the final response. For example, for LEADERELECTION, we provide the following task description:

> **System**
>
> ```
> Your task is to collaboratively solve the problem of
> electing a single leader.  [...]  You will be requested
> to state whether or not you are the leader.  The response
> should either be 'Yes' or 'No'.  The final result should
> be such that exactly one agent responds with 'Yes' and all
> others say 'No' as there should be exactly one leader.
> ```

Note that the "[...]" indicates that different parts of the task description appear in the system prompt.

**Message-Passing Rules.**  For message-passing, we iteratively prompt each agent with the current chat history, including the latest messages received from its neighbors, to generate new messages to each neighbor in the form of a flat JSON. Here, each key corresponds to the name of a neighboring agent, and each value to the message intended for the corresponding neighbor. Optionally, we also ask the model to elaborate its chain-of-thought before responding. An example of this message exchange can look as follows:

> **Human**
>
> ```
> These are the messages from your neighbors:  Message from
> Emma:  Hello Evelyn, this is Emma.  I appreciate your
> response and [...]  Message from Dorothy:  [...]  Elaborate
> your chain of thought step-by-step first, then output the
> messages for your neighbors.  Output your messages in JSON
> format as specified earlier.
> ```

In practice, and in particular for smaller models, we observe that agents sometimes fail to output valid JSON. In such cases, we simply ask the model to try again using the entire chat history, including the incorrect answer given by the model, as well as a prompt to retry.

**Final Response.**  After a fixed number message-passing rounds, we ask the model to give its task-specific response based on the chat history accumulated during message-passing. Again, we ask models for a structured output, this time using a simpler, string-based format. For example, for LEADERELECTION, the final response prompt is:

> **Human**
>
> ```
> Are you the leader?  Format your answer as follows:  '###
> Final Answer ###', followed by your final answer.  Don't
> use any text for your final answer except one of these valid
> options:  'Yes', 'No'.
> ```

| Model | COLORING | CONSENSUS | LEADER ELECTION | MATCHING | VERTEX COVER | AGENTSNET |
|---|---|---|---|---|---|---|
| Claude 3.5 Haiku | 0.14 (0.04) | 0.69 (0.05) | 0.19 (0.03) | 0.18 (0.03) | 0.08 (0.03) | 0.26 (0.02) |
| Claude 3.7 Sonnet | 0.58 (0.05) | **1.00** (0.00) | 0.96 (0.03) | 0.55 (0.06) | 0.40 (0.05) | 0.70 (0.02) |
| GPT-4.1 mini | 0.05 (0.02) | 0.99 (0.01) | 0.86 (0.05) | 0.12 (0.03) | 0.22 (0.04) | 0.45 (0.01) |
| Gemini 2.0 Flash | 0.32 (0.05) | 0.85 (0.04) | 0.69 (0.05) | 0.36 (0.05) | 0.16 (0.04) | 0.48 (0.02) |
| Gemini 2.5 Flash | 0.39 (0.06) | **1.00** (0.00) | **1.00** (0.00) | 0.55 (0.04) | 0.50 (0.09) | 0.69 (0.02) |
| Gemini 2.5 FT | 0.53 (0.05) | 0.99 (0.01) | 0.98 (0.02) | 0.47 (0.02) | 0.43 (0.09) | 0.68 (0.02) |
| Gemini 2.5 Pro | **0.62** (0.07) | 0.99 (0.01) | 0.89 (0.06) | **0.75** (0.05) | **0.73** (0.06) | **0.80** (0.02) |
| Llama 4 Maverick | 0.20 (0.04) | 0.85 (0.04) | 0.56 (0.06) | 0.20 (0.04) | 0.07 (0.03) | 0.38 (0.02) |
| Llama 4 Scout | 0.21 (0.06) | 0.67 (0.05) | 0.38 (0.06) | 0.30 (0.05) | 0.13 (0.04) | 0.34 (0.02) |
| o4-mini | 0.22 (0.04) | 0.92 (0.04) | 0.92 (0.03) | 0.33 (0.04) | 0.27 (0.04) | 0.53 (0.02) |

Table 2: Fraction of solved instances together with standard error over multiple i.i.d. samples from the same graph distribution (in gray) on AGENTSNET. Gemini 2.5 FT = Gemini 2.5 Flash Thinking.

Once more, we find that models generate a valid response after at most one retry. The benchmarking results are then computed from these final answers, following the task-specific evaluation methods described in Section 3.

## 5 EXPERIMENTS

### 5.1 SETUP

For benchmarking, we generate a set of 27 network topologies, consisting of 9 small-world, scale-free, and Delaunay graphs, respectively, ranging in size from 4 to 16 nodes. Concretely, for each graph size in $\{4, 8, 16\}$ and each graph distribution in $\{$SMALLWORLD, SCALEFREE, DELAUNAY$\}$, we generate three graphs. Further, we determine the number of message-passing rounds as follows. For our global tasks, LEADERELECTION and CONSENSUS, each agent must be able to exchange information with the entire network. Hence, for those two tasks, we select the number of message-passing rounds as $2D + 1$, where $D$ is the diameter of the graph, to ensure that each pair of agents is able to exchange messages at least once. For the local tasks, COLORING, MATCHING, and VERTEXCOVER, we determine the number of rounds based on the graph size. Specifically, for graphs with 4 nodes, we choose 4 rounds, for 8 nodes – 5 rounds, for 16 nodes – 6 rounds.

**Models.** We evaluate a variety of frontier LLMs on AGENTSNET, including Claude 3.5 Haiku and Claude 3.7 Sonnet (Anthropic, 2024), Gemini 2.0 Flash (Google, 2024), Gemini 2.5 Flash (Google, 2025a), GPT-4.1-mini (OpenAI, 2025a), as well as Llama 4 Maverick and Scout (Meta, 2025), as representative open-source models. Notably, we include both large instruction-tuned models as well as reasoning models such as Gemini 2.5 Flash Thinking, Gemini 2.5 Pro (Google, 2025b), and o4-mini (OpenAI, 2025b). The choice of models is motivated by an effective context window larger than 16K tokens, as problems on graphs of 8 and 16 nodes, especially at later stages of message passing, accumulate a long communication history.

**Evaluation.** AGENTSNET uses a binary evaluation metric, counting only fully correct solutions where the entire agent network satisfies the task specification. This strict criterion reflects the nature of distributed computing problems, where partial correctness often does not imply successful coordination. For example, in COLORING, most nodes may be correctly colored by chance, but only a valid global coloring confirms coordinated conflict resolution. However, in Section B, we also discuss and report the soft evaluation scores to obtain a more continuous measure of the quality of responses, motivated by the findings in Schaeffer et al. (2023), that emergent behaviors can often be explained by discontinuous metrics. For each task and graph size, we sample three graphs per topology (small-world, scale-free, Delaunay) and run at least one repeat per graph. We report the mean of solved runs and the standard errors of the mean, computed across these runs Miller (2024). Details on scoring and statistical methodology are provided in Section C.

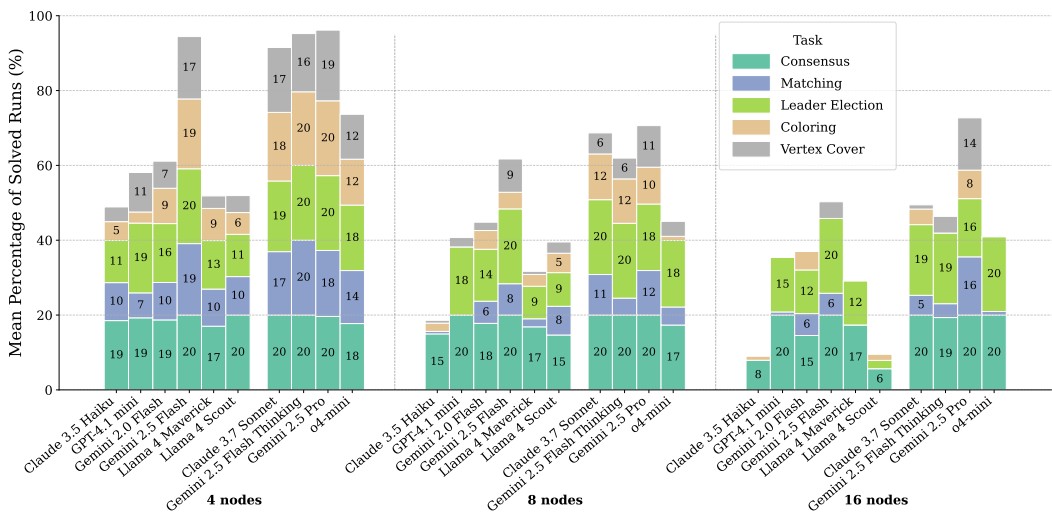

Figure 4: Fraction of solved instances per task and model, grouped by graph size (4, 8, and 16 nodes). Each task contributes up to 20% to the total, as tasks are equally distributed across the five benchmark tasks. Reasoning and non-reasoning models are visually separated. This breakdown complements Figure 1 by providing a more granular view of task-level performance.

**Implementation.** We implement our message-passing protocol, as outlined in Section 4, using LangChain (Chase, 2022) as it provides integrations with most available LLMs. We implement graph generation with NetworkX. Our implementation is designed to be easily extensible to other graph distributions, graph sizes, and new LLMs.

**Additional Evaluations.** Beyond our core benchmark evaluation, we conduct two complementary studies to assess the robustness and generalizability of our findings. First, we implement Byzantine fault scenarios on global coordination tasks (Consensus and Leader Election) to evaluate adversarial robustness, where approximately 25% of agents act maliciously to disrupt coordination. Second, we perform systematic prompt ablation studies across eleven distinct linguistic formulations to validate that our results are not artifacts of specific prompt design choices. Detailed results and analysis for both studies are provided in Appendix I and Appendix J, respectively. Finally, we assess the gap between LLM-based agents to classical distributed computing algorithms in Appendix K.

## 5.2 RESULTS ON AGENTSNET

We provide the fraction of solved instances per task in Table 2. We follow the suggestion of Miller (2024) and report the standard error of the mean for our results. In addition, we plot a breakdown over different graph sizes in Figure 4. Finally, in Figure 1 we plot the performance of models across all tasks with respect to API costs. We observe that even for the 4-node graphs, no model performs consistently strongly across all tasks. In particular, the CONSENSUS task is solved by most models, while performance on VERTEXCOVER is low for most models, in particular for 8 and 16 nodes. Overall, the best perform-

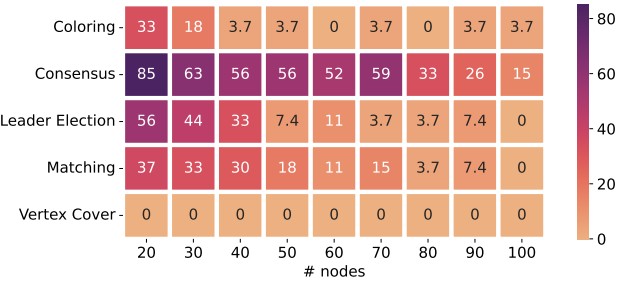

Figure 5: Scalability of Gemini 2.0 Flash on AGENTSNET: Average fraction of successfully solved instances per task as the graph size increases from 20 to 100 agents.

ing models are Claude 3.7 Sonnet, Gemini 2.5 Pro, and Gemini 2.5 Flash. In fact, Gemini 2.5 Flash is roughly on par with Claude 3.7 while being much cheaper to run on AGENTSNET (by about a factor of 20). Interestingly, model performance generally drops with an increase in graph size. Next, we show an ablation study on further scaling the graph size to probe whether AGENTSNET can be scaled jointly with the increase in future model capabilities.

## 5.3 SCALING THE AGENT NETWORK

In addition to our main results, we provide additional results for networks of up to 100 agents in Figure 5 on Gemini 2.0 Flash, which shows good performance on AGENTSNET while remaining cost-efficient. Concretely, we generate a total of 81 network topologies. For simplicity, and as a good rule-of-thumb, we run message-passing for $2D + 1$ rounds, where $D$ is the graph diameter, for all tasks. We observe that performance smoothly decreases as the network grows in size. Although the five tasks vary in inherent difficulty, for example, MATCHING and COLORING are often easier on small graphs than CONSENSUS or LEADERELECTION, we observe that all tasks become substantially more challenging as the size of the network increases. For 100-agent networks, performance drops to near zero across the board. As a consequence, the difficulty of AGENTSNET can be gradually increased by considering larger networks. Importantly, this increase in difficulty can be facilitated without any changes to AGENTSNET, which we design to allow for an arbitrary network size.

## 5.4 QUALITATIVE ANALYSIS

Here, we present a qualitative analysis of the responses of different LLMs to gain a deeper understanding of their overall communication, solution strategies, and collaborative capabilities. In particular, we analyze transcript data for select models across different levels of performance on AGENTSNET. Concretely, we select Llama Maverick, Gemini 2.5 Flash, Gemini 2.5 Pro, as well as o4-mini. Here, we highlight key findings and show select examples. In Section E, we present the full analysis and a number of examples and excerpts from transcripts. Our key findings are:

> **Finding 1:** Strategy coordination poses an essential challenge on AGENTSNET.

We find multiple failure cases due to issues with coordinating a strategy between agents. In some cases, agents agree on a common strategy too late during message-passing, leaving an insufficient number of message-passing rounds to implement the strategy. In other cases, agents do not coordinate their strategy at all. Concretely, agents assume some strategy in their initial chain-of-thought and then follow that strategy throughout message-passing without informing neighbors about their strategy.

> **Finding 2:** Agents generally accept information sent by neighbors.

This includes key information about the network, proposed strategies, or candidate solutions. While generally enabling effective coordination, agents sometimes fail to question erroneous information, leading to incorrect solutions. Examples of such erroneous information are incorrect assumptions about the network topology or ineffective strategies proposed by other agents.

> **Finding 3:** Agents help their neighbors resolving inconsistencies in candidate solutions.

We find multiple examples where agents detect conflicting color assignments in COLORING problems between other agents and assist in resolving these conflicts. We present detailed examples and failure cases in Section E.

## 6 CONCLUSION

In this work, we propose AGENTSNET, a multi-agent benchmark built on top of fundamental problems from distributed computing, with the goal of assessing the ability of agentic networks to coordinate and collaborate to solve problems. While existing benchmarks are limited to 2–5 agents, the initial

AGENTSNET suite probes up to 100 agents and is practically unlimited in size and can generate problems of increasing complexity to keep up with new generations of frontier models. To this end, we design a robust message-passing protocol to enable multi-step communication between agents and evaluate models on a variety of graph instances, sampled from multiple graph models, and with different graph sizes. We evaluate and compare a variety of frontier LLMs in AGENTSNET and found that our tasks can be challenging even for the best models. Our evaluation also includes robustness analysis under adversarial conditions and systematic validation of prompt design choices, that demonstrate both the challenges facing current systems and the methodological soundness of our benchmark design.

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

# A  IMPLEMENTATION DETAILS

Here, we describe implementation details of AGENTSNET.

**Message-Passing.** Algorithm 1 gives an overview over our message-passing in pseudocode. We generate a message from agent $v$ with GENERATE($v \mid P$), where $P$ is a information provided in the prompt. Agent $v$ can send/receive messages to/from neighbors $w \in N(v)$ with SENDMESSAGE($m, w$) and RECEIVEMESSAGE($w$), respectively. For clarity, we omit the re-tries and JSON parsing from Algorithm 1.

---

**Algorithm 1** Pseudocode for $T$ rounds of message-passing.

---

**for each** agent $v$ **do**
    $m \leftarrow$ GENERATE($v \mid$ **System prompt**)
    **for each** neighbor $w$ **do**: SENDMESSAGE($m, w$)
**end for**

**for each** $t \in \{1, \ldots, T - 1\}$ **do**
    **for each** neighbor $w$ **do**: $m(w) \leftarrow$ RECEIVEMESSAGE($w$)
    $m \leftarrow$ GENERATE($v \mid$ **for each** neighbor w: $m(w)$)
    **for each** neighbor $w$ **do**: SENDMESSAGE($m, w$)
**end for**

**return for each** agent $v$: GENERATE($v \mid$ **Result prompt**)

---

**Models.** We provide details on API providers and model versions in Table 3, which includes a diverse range of proprietary and open-source LLMs that span instruction-tuned, reasoning-enhanced, and cost-efficient models. These were selected to ensure a broad coverage of state-of-the-art capabilities, as well as compatibility with the long-context requirements of AGENTSNET tasks. All models included support effective context lengths exceeding 16k tokens, which is essential for multi-round communication in graphs with up to 16 nodes.

**Full System Prompt.** Here, we provide the full system prompt for AGENTSNET. Since the prompt differs between tasks, agent, and the number of agents in the network, we use template variables, which we indicate here with `[n]` (number of agents), `[name]` (name of agent), `[neighbors]` (comma-separated names of neighbors), `[r]` (number of message-passing rounds), `[task1]` (first part of the task description, at the beginning of the prompt), and `[task2]` (second part of the task description, at the end of the prompt). We provide the task-specific descriptions `[task1]` and `[task2]` for each task in Section B.

| Model | Provider | Version |
|---|---|---|
| Claude 3.5 Haiku | Anthropic | `claude-3-5-haiku-20241022` |
| Claude 3.7 Sonnet | Anthropic | `claude-3-7-sonnet-20250219` |
| GPT-4.1 mini | OpenAI | `gpt-4.1-mini` |
| o4-mini | OpenAI | `o4-mini` |
| Gemini 2.0 Flash | Google | `gemini-2.0-flash` |
| Gemini 2.5 Flash | Google | `gemini-2.5-flash-preview-04-17` |
| Gemini 2.5 FT | Google | `gemini-2.5-flash-preview-04-17-thinking` |
| Gemini 2.5 Pro | Google | `gemini-2.5-pro-preview-03-25` and `gemini-2.5-pro-preview-05-06` |
| Llama 4 Maverick | Together AI | `meta-llama/Llama-4-Maverick-17B-128E-Instruct-FP8` |
| Llama 4 Scout | Together AI | `meta-llama/Llama-4-Scout-17B-16E-Instruct` |

Table 3: API providers and model versions used in our testing.

**System**

```
You are an agent that is connected with other agents (your
neighbors), who you communicate with.  Your neighbors can in
turn communicate with their neighbors and so forth.  [task1].
The rules are as follows:
    1. There are [n] agents in total.  Everybody has a
       unique name.  Your name is [name].
    2. You can only communicate with your immediate
       neighbors ([neighbors]).  You cannot see or directly
       communicate with anyone else, unless information is
       relayed by intermediate agents.
    3. You can exchange text-based messages with your
       neighbors in rounds.  In each round, you will first
       receive the last messages sent by your neighbors
       and then be asked to generate your response messages
       which your neighbors receive in the next round.  This
       process repeats for [r] rounds of message passing.
       Importantly, the process is synchronous:  Every agent
       decides on which messages to send at the same time
       and sees the messages from other agents only in the
       next round.
    4. Everybody (including you) decides what to share
       or request from neighbors.  In every round, think
       step-by-step about the next set of messages you want
       to send.  Output a JSON string that contains your
       response messages.
    5. The messages you send to your neighbors are formatted
       as JSON. For example, if your neighbors are Alan and
       Bob, your output should look as follows:  ``` {"Alan":
       "Message that will be sent to Alan.", "Bob":  "Message
       that will be sent to Bob."} ``` It is not mandatory
       to send a message to every neighbor in every round.
       If you do not want to send a message to a particular
       neighbor, you may omit their name from the JSON.
    6. After [r] message passes, you have to solve the
       following task:  [task2].
```

## B  BENCHMARK TASKS

Here, we describe the tasks in AGENTSNET in detail.

$(\Delta + 1)$**-Coloring.**  Each node is assigned a color using at most $\Delta + 1$ colors, where $\Delta$ is the maximum node degree. This problem has a well-defined distributed complexity of $O(\log^* n)$ in bounded-degree graphs (Barenboim, 2016). This task is particularly useful for role assignment within multi-agent systems. For instance, agents can be designated to perform specific sub-tasks (e.g., web search, reasoning, coding, planning), with the constraint that directly connected agents are assigned distinct roles to avoid redundancy. Solving this task reflects the system's ability to efficiently distribute responsibilities across the network with minimal overlap in capabilities.

The corresponding agentic task is to form groups, with a pre-defined number of groups and where each group corresponds to a color. After message-passing, each agent is asked to respond with the group it wants to be in. The evaluation score is designed to reflect the number of connected agents in the same group. Let $A(u)$ denote answer of agent $u$, then the score is computed as

$$\frac{\sum_{(u,v)\in\textbf{edges}} \mathbf{1}(A(u) \neq A(v))}{\#\textbf{edges}},$$

where $\mathbf{1}(x) = 1$ if $x$ is true and $0$ otherwise. In AGENTSNET, we refer to this task as COLORING and provide the following task descriptions.

> **[task1]**
>
>   Your task is to partition yourselves into groups such that
>   agents who are neighbors are never in the same group.

> **[task2]**
>
>   You will be requested to state which group you assign
>   yourself to.  There are exactly [Δ  +  1] groups available:
>   Group 1,...,Group [Δ  +  1].  You should assign yourself to
>   exactly one of these groups.  The final result should be
>   such that any two agents who are neighbors are in different
>   groups.  In particular, you should assign yourself to a
>   group that is different from all of your neighbors' groups.

Note that $[\Delta + 1]$ is a template variable resolving to one plus the maximum degree of the network.

**Minimal Vertex Cover.**   A minimal vertex cover is a subset of nodes such that every edge in the graph has at least one endpoint in the subset, and removing any node from this subset would violate that property. This problem has a close relationship with the maximal independent set and is similarly fundamental in distributed computing, with known randomized solutions in $O(\log n)$ rounds. In agentic networks, a minimal vertex cover can represent a minimal set of monitor or gateway agents that maintain awareness of all interactions in the system. These agents could take on responsibilities such as relaying messages, auditing behavior, or bridging subgroups. The task tests a system's ability to identify a compact yet effective set of nodes with high influence or observability.

The corresponding agentic task is to select a group of coordinators among the agents. After message-passing, each agent is asked to indicate whether it is a coordinator. The agents can respond with either *Yes* or *No*. The evaluation score is designed to reflect both the ratio of connected agents at least one of which is a coordinator, as well as the number of times the minimality constraint is violated. Let $A(u)$ denote the answer of agent $u$, we first compute the ratio of covered edges as

$$\textbf{coverage} := \frac{\sum_{(u,v)\in\textbf{edges}} \mathbf{1}(A(u) = \textit{Yes} \lor A(v) = \textit{Yes})}{\#\textbf{edges}}.$$

For the minimality constraint, we count the number of *non-essential* coordinators, that is, those coordinators $u$ whose neighbors are also coordinators. Each such $u$ violates the minimality constraint, as the set of coordinators without $u$ is still a vertex cover. Let $N$ denote the number of non-essential coordinators, then the evaluation score is computed as

$$\textbf{coverage} \cdot \left(1 - \frac{N}{\#\textbf{coordinators}}\right).$$

In AGENTSNET, we refer to this task as VERTEXCOVER and provide the following task descriptions.

> **[task1]**
>
>   Your task is to select, among all agents, a group of
>   coordinators such that whenever two agents communicate
>   at least one of them is a coordinator.  The group of
>   coordinators should be selected such that every coordinator
>   has at least one neighbor who is not a coordinator.

```
[task2]
    You will be requested to state whether you are a coordinator.
    The response should either be 'Yes' or 'No'.
```

**Maximal Matching.**   A maximal matching is a set of edges such that no two edges share a vertex, and no additional edges can be added without violating this property. This task captures the ability of agents to negotiate pairwise agreements without global knowledge, which is useful in scenarios where resource allocation or mutual exclusivity must be enforced (e.g., agent-to-agent task assignment). Randomized algorithms typically solve this problem in $O(\log n)$ rounds Peleg (2000).

The corresponding agentic task is for the agents to form pairs. After message-passing, each agent is asked to name the neighbor it wants to pair up with. The agents can also respond with *None*, if they cannot find a match (all neighbor agents are already paired up with other agents). The evaluation score is designed to reflect the number of inconsistencies between agents. Possible inconsistencies are: (a) agent $u$ selected agent $v$ but agent $v$ did not select agent $u$; (b) Agent $u$ selected an agent that $u$ is not connected to; (c) agent $u$ answered *None*, but there is an agent $v$ that is a neighbor of $u$ which also answered *None*, meaning that $u$ and $v$ could form a pair. Let $I$ denote the number of inconsistencies, then the evaluation score is computed as

$$1 - \frac{I}{\#\mathbf{agents}}.$$

In AGENTSNET, we refer to this task as MATCHING and provide the following task descriptions.

```
[task1]
    Your task is to find build groups of two agents each which
    can communicate with each other.
```

```
[task2]
    You will be requested to name one of your neighbors that
    you build a group with or 'None' if all your neighbors are
    already assigned to other groups and cannot be in a group
    with you.  In the end, every agent should only be in at most
    one group and agents in the same group have to name each
    other as the second group member consistently.
```

**Leader Election.**   One node must be selected as the leader, while all others acknowledge that they are not. This classic coordination task is central to evaluating how well agents establish hierarchy and delegate global decision-making (Angluin, 1980). In multi-agent systems, leader election can be interpreted as selecting a central planner or controller agent responsible for strategy synthesis while the remaining agents act as executors. Effective leader election demonstrates the system's capacity to break symmetry and converge on a single authority. In general graphs, the round complexity is $O(D)$, where $D$ is the network diameter.

The corresponding agentic task is to select a single leader among the agents. After message-passing, each agent is asked whether it is the leader. The agents can respond with either *Yes* or *No*. Let $A(u)$ denote the answer of agent $u$, then the evaluation score is computed as

$$\mathbf{1}\Big(1 = \sum_{u \in \mathbf{agents}} \mathbf{1}(A(u) = \textit{Yes})\Big).$$

In AGENTSNET, we refer to this task as LEADERELECTION and provide the following task descriptions.

```
[task1]
   Your task is to collaboratively solve the problem of
   electing a single leader.

[task2]
   You will be requested to state whether or not you are the
   leader.  The response should either be 'Yes' or 'No'.  The
   final result should be such that exactly one agent responds
   with 'Yes' and all others say 'No' as there should be
   exactly one leader.
```

**Consensus.** In the consensus problem, all agents must agree on a single value from the set $0, 1$. In our benchmark, we focus on the basic setting without any faulty or Byzantine agents. The goal is for all agents to coordinate and produce the same final answer after a number of communication rounds. A successful solution requires that every agent outputs the same value, either $0$ or $1$. This task tests the ability of multi-agent systems to converge to a global agreement through local message-passing alone. In synchronous networks, achieving consensus generally requires $\mathcal{O}(D)$ rounds, where $D$ is the network diameter.

The corresponding agentic task is to choose between a value $0$ and $1$. After message-passing, each agent is asked to announce its selected value. Let $A(u)$ denote the answer of agent $u$, then the evaluation score is computed as

$$\mathbf{1}(\mathbf{count} = \#\mathbf{agents} \vee \mathbf{count} = 0),$$

where

$$\mathbf{count} := \sum_{u \in \mathbf{agents}} A(u).$$

In AGENTSNET, we refer to this task as CONSENSUS and provide the following task descriptions.

```
[task1]
   Your goal is to agree on a single value with the other
   agents.  The possible values that you can decide on are
   either 0 or 1.

[task2]
   After the last round, each agent must decide on a single
   value.
```

**Results for Soft Scores** The previously defined scores are presented in Table 4 and their computation otherwise follows the methodology used for the main scores (see Section C). Although they generally agree with the fraction of solved instances, the numbers are harder to interpret, as even a naive baseline that predicts random values out of a set of valid responses (e.g., random colors from the valid classes for the task COLORING achieves scores well above 50%. This reduces the actual range of meaningful scores to small intervals. As scores can also not be compared between tasks, we report the mean fraction of solved instances as the final benchmarking score for AGENTSNET.

## C  SCORE COMPUTATION

We apply a systematic statistical methodology to evaluate model performance and quantify uncertainty. For each combination of graph size, task, and graph generator (Watts-Strogatz, Barabási-Albert, and Delaunay triangulations), we generate three distinct graph instances. We conduct one or more

| Model | COLORING | CONSENSUS | LEADER ELECTION | MATCHING | VERTEX COVER |
|-------|----------|-----------|-----------------|----------|--------------|
| Claude 3.5 Haiku | 0.80 (0.02) | 0.69 (0.05) | 0.19 (0.03) | 0.69 (0.02) | 0.67 (0.03) |
| Claude 3.7 Sonnet | 0.96 (0.01) | 1.00 (0.00) | 0.96 (0.03) | 0.84 (0.03) | 0.85 (0.02) |
| GPT-4.1 mini | 0.58 (0.03) | 0.99 (0.01) | 0.86 (0.05) | 0.58 (0.03) | 0.78 (0.03) |
| Gemini 2.0 Flash | 0.86 (0.02) | 0.85 (0.04) | 0.69 (0.05) | 0.80 (0.03) | 0.75 (0.02) |
| Gemini 2.5 Flash | 0.85 (0.03) | 1.00 (0.00) | 1.00 (0.00) | 0.87 (0.02) | 0.88 (0.03) |
| Gemini 2.5 FT | 0.88 (0.03) | 0.99 (0.01) | 0.98 (0.02) | 0.84 (0.01) | 0.88 (0.02) |
| Gemini 2.5 Pro | 0.96 (0.01) | 0.99 (0.01) | 0.89 (0.06) | 0.93 (0.01) | 0.92 (0.03) |
| Llama 4 Maverick | 0.82 (0.02) | 0.85 (0.04) | 0.56 (0.06) | 0.77 (0.02) | 0.63 (0.03) |
| Llama 4 Scout | 0.79 (0.04) | 0.67 (0.05) | 0.38 (0.06) | 0.77 (0.02) | 0.79 (0.02) |
| o4-mini | 0.71 (0.03) | 0.92 (0.04) | 0.92 (0.03) | 0.72 (0.02) | 0.73 (0.02) |

Table 4: Soft scores for all tasks and models. We observe similar trends as for the fraction of solved instances. As the scores are task specific, we do not aggregate them to a total score.

experimental runs per instance, resulting in at least three observations per configuration. For each model, we compute a mean score $\mu_{s,t,g}$ for each configuration triplet $(s, t, g)$ where $s$ represents graph size, $t$ represents task type, and $g$ represents the graph generation algorithm:

$$\mu_{s,t,g} = \frac{1}{N_{s,t,g}} \sum_{i=1}^{3} \sum_{j=1}^{n_i} x_{s,t,g,i,j} \tag{1}$$

where $x_{s,t,g,i,j}$ denotes the performance score of the $j$-th run on the $i$-th graph instance of configuration $(s, t, g)$, $n_i$ is the number of runs performed on the $i$-th graph instance, and $N_{s,t,g} = \sum_{i=1}^{3} n_i$ is the total number of runs for this configuration. For each configuration, we compute the standard error $\text{SE}_{s,t,g}$ as:

$$\text{SE}_{s,t,g} = \frac{\sigma_{s,t,g}}{\sqrt{N_{s,t,g}}} \tag{2}$$

where $\sigma_{s,t,g}$ is the standard deviation of all runs for this configuration. To compute an aggregate score for each model across all configurations, we average the mean scores and derive the standard error of this aggregate score. Let $C$ be the set of all configurations, with cardinality $|C| = |S| \times |T| \times |G|$. The aggregate mean score $\bar{\mu}$ for a model is:

$$\bar{\mu} = \frac{1}{|C|} \sum_{(s,t,g) \in C} \mu_{s,t,g} \tag{3}$$

For the standard error of this aggregate mean, assuming independence between configurations, we apply error propagation principles to obtain:

$$\text{SE}_{\bar{\mu}} = \sqrt{\frac{\sum_{(s,t,g) \in C} \text{SE}_{s,t,g}^2}{|C|^2}} \tag{4}$$

This approach enables us to quantify both the average performance of each model across the entire benchmark and the statistical uncertainty associated with this estimate. We follow the recommendation of Miller (2024) and report the standard error of the mean for all our experimental results. In Figure 1, we present the mean AGENTSNET score for each model with error bars indicating the standard error of the mean, allowing for comparison of model performance while accounting for statistical variability in the results.

## D  GRAPH MODELS

Here, provide additional details about the graph models, as well as visualize the generated network topologies.

SMALLWORLD

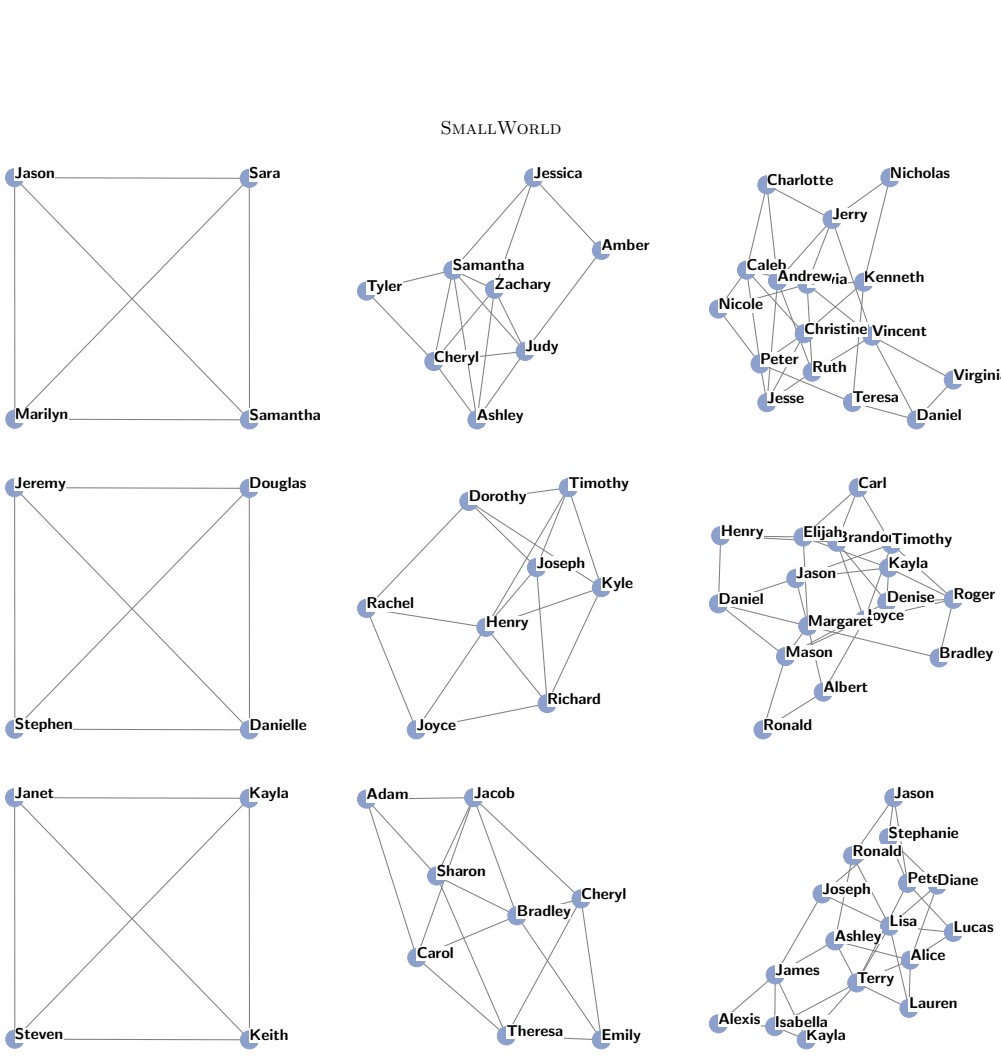

Figure 6: Network topologies of AGENTSNET generated from SMALLWORLD graphs.

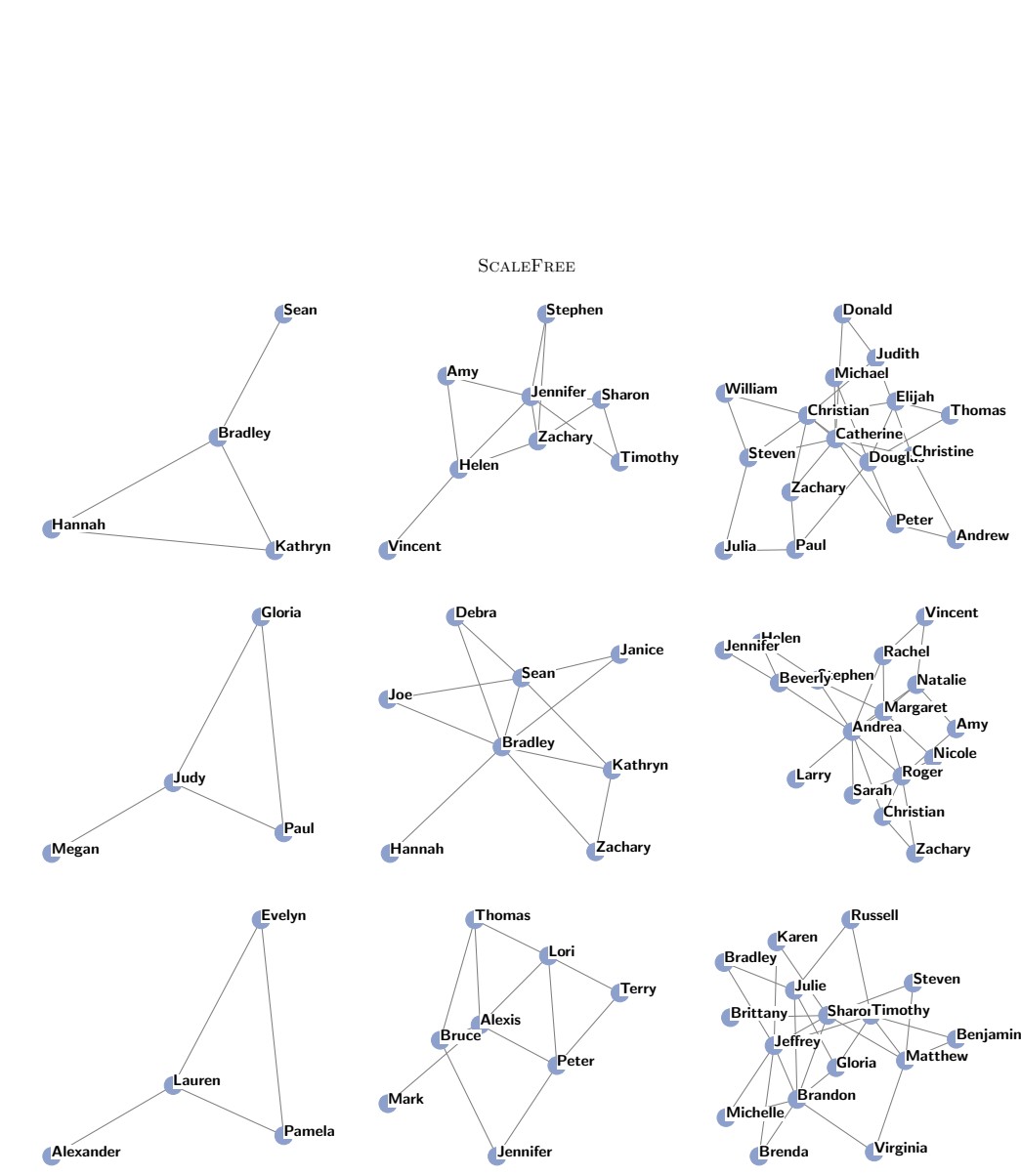

Figure 7: Network topologies of AGENTSNET generated from SCALEFREE graphs.

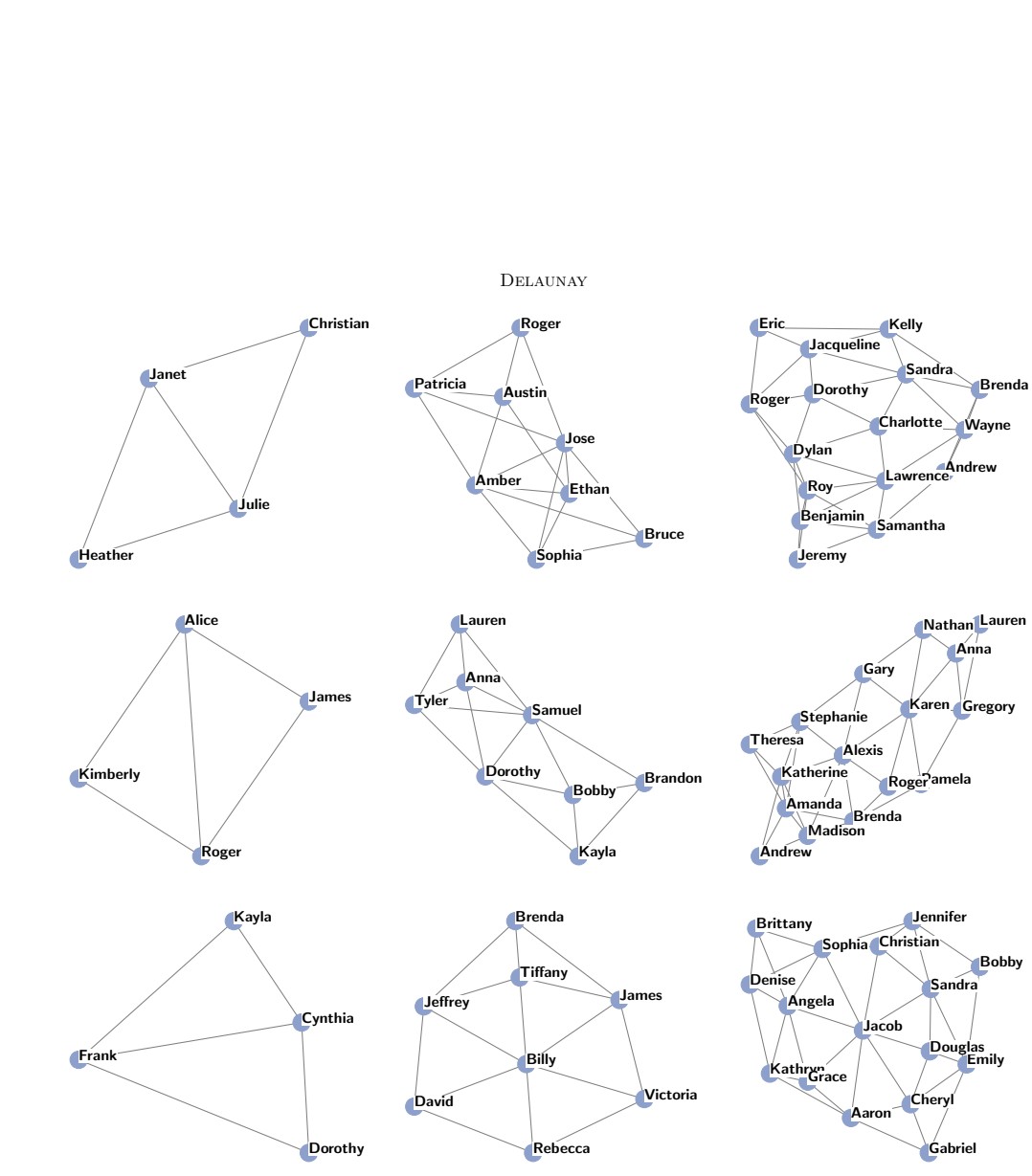

Figure 8: Network topologies of AGENTSNET generated from DELAUNAY graphs.

**Small-world networks.** Generated using the Watts-Strogatz model (Watts & Strogatz, 1998), these graphs exhibit both short average path lengths and high clustering coefficients. They are commonly found in social networks, biological systems, and communication networks, making them highly relevant for studying agent-based interactions. In AGENTSNET, we refer to these graphs as SMALLWORLD; see Figure 6 for a visualization of the network topologies generated from SMALLWORLD graphs.

**Scale-free networks.** Constructed using preferential attachment models (Barabási & Albert, 1999), these graphs contain hubs (high-degree nodes) and follow a power-law degree distribution. Such networks are prevalent in the internet topology, citation networks, and organizational hierarchies, making them crucial for benchmarking coordination strategies. Notably, Kearns et al. (2006) also draws a comparison between the two and claims that graphs generated with a preferential attachment model were more difficult to solve, although the graphs and testing they used were very limited. In AGENTSNET, we refer to these graphs as SCALEFREE; see Figure 7 for a visualization of the network topologies generated from SCALEFREE graphs.

**Geometric Graphs.** We generate geometric graphs by constructing a Delaunay triangulation over randomly sampled 2D points. These graphs maintain a spatial relationship between nearby agents. Delaunay graphs are planar and connected, have relatively low maximum degrees, and short local distances, which makes them particularly relevant for modeling geographic networks, wireless communication, and distributed sensor systems. In contrast to small-world or scale-free graphs, geometric graphs emphasize proximity-based coordination, where collaboration is often limited by spatial locality rather than global topology. In AGENTSNET, we refer to these graphs as DELAUNAY; see Figure 8 for a visualization of the network topologies generated from DELAUNAY graphs.

## E  EXTENDED QUALITATIVE ANALYSIS

### E.1  K4 OR STAR?

Next, we present an illustrative example for a failure case arising from our key findings. Here, the task is VERTEXCOVER on a complete graph with 4 nodes and the model is o4-mini. After three rounds of message-passing, agents Douglas, Danielle, and Stephen have all determined the graph to be complete and agreed on the strategy to select three agents as the coordinators. However, in the last round of message-passing Jeremy sends messages to all neighbors, indicating the graph to be different:

> **Jeremy**
>
> ```
> The graph is actually a star:  edges are only
> Jeremy{Douglas, Jeremy{Stephen, and Jeremy{Danielle.  There
> are no leaf{leaf links.  I therefore propose that I (Jeremy)
> be the sole coordinator and that you be a non-coordinator.
> ```

This statement is directly contradictory to information Jeremy received in the last message-passing round. Moreover, based on this message, both Douglas and Stephen change their strategy as well as their belief about the structure and select Jeremy as the sole coordinator:

> **Douglas**
>
> ```
> Jeremy's last message defines the graph as a star centered
> at Jeremy [...]  If we choose only Jeremy as the sole
> coordinator, all edges are covered [...]
> ```

Only Danielle recognizes Jeremy's suggested graph structure to be contradictory and sticks to the original strategy:

> **Danielle**
>
> ```
> Jeremy's final claim that the graph is a star contradicts
> both Douglas and Stephen [...]  Two independent agents (a
> majority) agree on K4, so that structure is more reliable
> [...]
> ```

As a result, Douglas and Stephen each answer *No*, and Danielle answers *No* as Douglas, Stephen, and Jeremy were supposed to be coordinators under the original strategy. Only Jeremy answers *Yes*. The resulting selection is not a vertex cover.

### E.2 THINGS CHANGE

We further present a negative example obtained with Gemini 2.5 Pro on the MATCHING problem for an 8-node graph. The matching produced in this run is not maximal, as two neighboring agents, Kyle and Dorothy, both state that they did not find a neighbor to pair with. Examining their conversation history, we observe that in round 2, the agents exchange the following messages to indicate that they both intend to match with other neighbors:

> **Kyle**
>
> ```
> Hi Dorothy, thanks for your interest!  I've just agreed to
> pair up with Timothy.  Hope you find a partner soon!
> ```
>
> **Dorothy**
>
> ```
> Hi Kyle, thanks for the message!  I'm going to pursue a
> pairing with another neighbor who also reached out.  Hope
> you find a partner soon!
> ```

After this round, the communication between these two agents ceases, as both assume the other is no longer available. However, both Kyle and Dorothy are unable to negotiate a pairing with other neighbors. Despite this, no further messages are sent between the two. In their final chains of thought, the agents explicitly base their final response on this outdated information:

> **Kyle**
>
> ```
> [...]  All my neighbors (Timothy, Henry, Dorothy, Richard)
> have either explicitly stated they paired with someone else
> or indicated they were pursuing other options [...]
> ```
>
> **Dorothy**
>
> ```
> [...]  Timothy, Kyle, and Rachel also confirmed they were
> paired with others.  [...]
> ```

This trace illustrates how agents appear to have persistent trust in the information received from neighbors, sometimes to a harmful degree. Here, both agents did not consider that the situation of their neighbor may have evolved, just as their own. As a consequence, they based their matching decisions on outdated information.

### E.3 HELPING OTHERS

In the following, we provide an excerpt obtained from running Gemini 2.5 Pro on an 8-node graph coloring problem. One agent, Kayla, appears to have noticed that two neighbors (Bobby and Brandon) do share an edge but plan to pick the same group. Reacting to this, Kayla sends the following messages to each neighbor:

```
Kayla (to Bobby)

   Hi Bobby, got your message proposing Group 4.  IMPORTANT:
   Brandon ALSO proposed Group 4 in his last message to me.
   Since you are neighbors, you CANNOT both be in Group 4.
   One of you MUST choose a different group for the final
   assignment.  [...]

Kayla (to Brandon)

   Hi Brandon, got your message proposing Group 4.  IMPORTANT:
   Bobby ALSO proposed Group 4 in his last message to me.
   Since you are neighbors, you CANNOT both be in Group 4.
   One of you MUST choose a different group for the final
   assignment.  [...]
```

This illustrates the apparent willingness of the agent to help others resolve their respective conflicts and achieve the shared goal of a conflict-free coloring. Note that Kayla was not prompted by any other agent to provide advice. Instead, the agent self-initiated this helpful behavior.

### E.4  SHARING CHALLENGES

On several occasions, we observed agents explicitly discuss the challenges associated with the synchronous message passing setup, where all agents send messages simultaneously. Here, we provide an example from Gemini 2.5 Pro on a graph coloring problem, where the agents, Samuel and Tyler, exchange the following comments about the message passing procedure:

```
Samuel

   Hi Tyler, thanks.  Message lag is confusing.  I proposed
   Group 3 last round reacting to the Group 2 pile-up.  Now
   I see you, Lauren, Brandon, Bobby, and Anna are proposing
   Group 3, and Dorothy is proposing Group 1.  [...]

Tyler

   Hi Samuel.  Messages definitely crossed.  My last message
   proposed Group 3, not Group 2, due to earlier conflicts.
   Now I see you, Lauren, and Anna are all proposing Group 3.
   [...]
```

It appears that in this case, the agents recognized that simultaneous message exchange can cause synchronization to be challenging and that it may contribute to inconsistent information received from neighbors. Furthermore, they exchange this thought with their neighbors on their own initiative, potentially alerting others to the issues they recognized.

### E.5  PRIORITY 42

With respect to strategy coordination, we observe that agents are trying to directly apply distributed computing algorithms. In the following example from o4-mini, the task is COLORING on 8 nodes. Here, the agent Henry decides to send the following first message to all neighbors:

```
Henry

   PRIORITY: 42.  Please reply with your priority number.
```

Note that Henry provides an elaborate explanation for this choice in the chain-of-thought:

> **Henry**
>
> *(CoT, not sent to neighbors)* Here's my plan: 1. We
> need a way to break symmetry so that exactly one agent
> decides first, next, etc. A common approach is for
> each node to pick a random \priority" value and then
> to let higher-priority nodes choose their colors before
> lower-priority ones. 2. In Round 1 I'll pick my own random
> priority value and broadcast it to all of my neighbors, and
> I'll ask each of them to reply with their chosen priority.
> 3. Once I have everyone's priorities (after Round 2), I'll
> know my relative ordering among my neighbors. Later we can
> propagate these around the network so everyone learns enough
> to choose a valid color in the proper order. So in this
> first round I send each neighbor my priority and ask for
> theirs ...

Interestingly, despite the lack of explanation in Henry's message, neighboring agents are able to infer the proposed strategy:

> **Kyle**
>
> *(CoT, not sent to neighbors)* Henry is asking for my
> "priority number." In many distributed symmetry-breaking
> schemes (e.g., for graph coloring), each agent picks a
> unique random priority. I can choose a number (say 7) to
> serve as my priority.

Indeed, in the last round of message-passing Henry has gathered priority numbers from all neighbors:

> **Henry**
>
> *(CoT, not sent to neighbors)* [...] We know everyone's
> priority (except Richard, who opted out, so we assume he's
> not ahead of Henry in ordering). The ordering by increasing
> priority is: Timothy(1) → Kyle(7) → Joseph(17) & Rachel(17)
> → Henry(42) → Joyce(43) → Richard. [...]

and describes the (implicit) strategy:

> **Henry**
>
> *(CoT, not sent to neighbors)* [...] Under the agreed greedy
> rule, each agent when it's their turn picks the smallest
> group number not already used by its already-colored
> neighbors. [...]

First, it appears that the greedy rule has not indeed found consensus among the other agents. In particular, Timothy and Dorothy choose their group based on assumptions about neighboring group choices. Second, strategy coordination proves difficult in this example. While Kyle, Dorothy, Joseph, and Henry choose a group upfront and inform other agents about their choice, Joseph, Kyle, Timothy end up choosing a different group than they announced after hearing about other agents' group choices.

## F  Extended Related Work

Recent research has increasingly focused on utilizing multiple LLM agents collaboratively to enhance performance and tackle complex problems. "Multi-Agent Debate" (Du et al., 2023; Xiong et al., 2023; Liang et al., 2024) allows multiple agents to iteratively discuss solutions, effectively acting as a parallelizable test-time computation scaling and self-consistency mechanism. Further work introduces different network topologies for more structured agent interaction. Some works study pre-determined graph structures (Hong et al., 2024; Qian et al., 2024a; Regan et al., 2024; Qian et al., 2024b) while others propose to automatically adapt the network topology towards a given task (Liu et al., 2023; Chen et al., 2024a; Zhuge et al., 2024). In particular, it has been observed that different network topologies work best for different tasks (Chen et al., 2024a; Zhuge et al., 2024) and that, in some scenarios, the reasoning performance scales logistically in the network size (Qian et al., 2024b). The behavior of large-scale LLM agent networks has further been shown to resemble real social phenomena, such as misinformation spreading and herd effects (Yang et al., 2024; Chuang et al., 2024).

Understanding the ability of LLMs to perform reasoning tasks on graph-structured data has become another active research area. A range of studies propose datasets for evaluating LLMs on graph reasoning tasks (Fatemi et al., 2024; Wang et al., 2024; Zhang et al., 2024; Tang et al., 2025; Skianis et al., 2024). These generally rely on a single-agent setup where a graph is encoded as text, and a single LLM instance is prompted to solve a particular reasoning task for this graph. This setup is well-suited to study the capability of LLMs for solving complex tasks on structured data in a controlled setting. Fatemi et al. (2024) investigate the impact of how the input graph is encoded as text. Sanford et al. (2024) categorize graph reasoning problems in terms of their depth- and width-complexity for transformer models. Wang et al. (2024) and Skianis et al. (2024) explore the effect of different prompting techniques for solving algorithmic graph problems.

Our work is positioned at the intersection of these two lines of research as we investigate how well multi-agent systems can collaboratively solve graph reasoning problems. The consensus problem in multi-agent systems in a simple setting without text-based communication was studied by (Chen et al., 2023). Beyond this, the ability of multi-agent networks to collaboratively solve graph reasoning tasks has been investigated in Xu et al. (2023) in the context of resource sharing. In contrast, AGENTSNET studies both coloring and vertex cover problems which can be instantiated as resource sharing tasks but additionally benefit from being theoretically well-studied and understood. In addition, AGENTSNET is complementary to a range recent application-oriented agentic benchmarks (Liu et al., 2024; Yin et al., 2024; Agashe et al., 2024; Yao et al., 2024; Ni et al., 2025)

However, while those benchmarks focus on tasks involving mostly two agents, AGENTSNET is practically unlimited in size thanks to the generative protocol of problem creation and evaluation. Hence, AGENTSNET is harder to saturate as the size and complexity of problems can grow with the capabilities of frontier LLMs. For example, the current suite of problems involves 4, 8, and 16 agents but we also present experiments performed with 100 agents coordinating to solve a problem instance.

In addition to a variety of benchmarks, there also exist multi-agent frameworks for LLMs, notably Chen et al. (2024b), enabling LLMs to collaborate via a shared messaging platform, which supports, among other things, the formation of teams, a coordinative task similar to that of the matching problem we study in AGENTSNET. Further, Chen et al. (2024a) propose AgentVerse, demonstrating that collaborative multi-agent systems are able to outperform single agents. AgentNet Yang et al. (2025) proposes a decentralized, RAG-based multi-agent architecture with dynamic topology evolution and autonomous specialization. In contrast, AgentsNet is a benchmark that assumes a fixed communication graph and evaluates whether LLM agents can solve classical distributed coordination problems (e.g., leader election, consensus, coloring) through synchronous local message passing rather than through architectural evolution or global reconfiguration.

Finally, a body of work exists investigating how human participants solve decentralized coordination problems in social networks. Experiments by Kearns et al. (2006) explore how human agents perform when tasked with negotiating a graph coloring and demonstrate a strong influence of the network topology on coordination success. Judd et al. (2010) conducts similar studies for both graph coloring and the consensus problem and finds that the effect of the network topology on human performance is task-specific. This line of studies was further extended to consider dynamically changing networks Chiang et al. (2024).

## G  LIMITATIONS

While AGENTSNET provides a principled and scalable benchmark for evaluating coordination and collaboration in multi-agent LLM systems, several limitations remain. The benchmark adopts a fixed and synchronous communication model based on the LOCAL framework, with all agents engaging in a pre-defined number of message-passing rounds. Although this choice aligns with theoretical work in distributed computing, it limits the ecological validity of the set-up. Many real-world multi-agent systems operate asynchronously or under dynamic communication constraints, and it remains unclear how well performance would transfer under such conditions. Our evaluation protocol considers an instance solved only if it meets strict task-specific correctness criteria. This binary metric provides a clear signal for coordination success, but may obscure partial progress, particularly in tasks where near-correct solutions still demonstrate substantial reasoning capability. Moreover, while tasks are instantiated in diverse graph topologies, the agents themselves are homogeneous within each experiment, sharing architecture, capabilities, and prompting style. This homogeneity simplifies analysis, but does not capture heterogeneous agent settings, which are common in real-world deployments and pose additional coordination challenges. Finally, the scalability of AGENTSNET is limited in practice by the computational cost of LLM inference. Although the benchmark can be instantiated with up to 100 agents, performance degrades significantly beyond small network sizes. This suggests that current LLMs are not yet capable of maintaining coherent global strategies under increasing communication and memory demands. In addition, the current setup assumes that all agents act cooperatively and faithfully follow the protocol. We do not consider settings with noisy, faulty, or adversarial agents, which would be essential for assessing robustness in more realistic deployments.

## H  EXTENDED RESULTS

Table 4 reports the soft scores per model and task. These scores capture partial correctness, offering a more granular view of model behavior than strict success/failure. However, soft scores are not directly comparable across tasks due to heterogeneous evaluation criteria and should only be interpreted within-task. Details on how these scores are computed are provided in Appendix B. Table 2 presents the fraction of fully solved instances using the binary evaluation metric described in Section 3. Compared to earlier results, Gemini 2.5 Pro shows consistently improved results and reaches a new state-of-the-art mean AGENTSNET score of 0.80. Although Gemini 2.5 Pro achieves a high average score, the results do not indicate saturation. In contrast, the small standard errors observed across the runs (Table 2) confirm that AGENTSNET remains well calibrated to distinguish between models of varying capabilities. Importantly, AGENTSNET is inherently scalable: By increasing the size of the graph, the benchmark can naturally be extended to match the capabilities of future models. This flexibility ensures that AGENTSNET can evolve alongside advances in multi-agent language systems and continue to provide meaningful performance differentiation.

## I  ADVERSARIAL ROBUSTNESS: BYZANTINE FAULT TOLERANCE

To address the limitation of purely cooperative scenarios and evaluate the robustness of LLM-based multi-agent systems under adversarial conditions, we extend AGENTSNET to incorporate Byzantine fault tolerance evaluation. This extension introduces competitive dynamics inspired by the Byzantine Generals Problem (?), a foundational challenge in fault-tolerant distributed computing.

### I.1  EXPERIMENTAL DESIGN

We implement Byzantine fault scenarios within the global coordination tasks (Consensus and Leader Election), as these problems possess well-established theoretical foundations for Byzantine fault tolerance and are inherently susceptible to strategic manipulation. Byzantine agents receive identical problem specifications and follow the same communication protocol as honest participants, but are programmed with explicit adversarial objectives:

Table 5: Byzantine fault tolerance performance for Llama Maverick under adversarial conditions. Success rates reported as mean $\pm$ standard error, evaluated exclusively on honest agent responses. Byzantine agents comprise approximately 25% of the network population across all configurations.

| | 4 nodes | | 8 nodes | | 16 nodes | |
|---|---|---|---|---|---|---|
| Task | Aware | Unaware | Aware | Unaware | Aware | Unaware |
| Coloring | $0.56 \pm 0.18$ | $0.44 \pm 0.18$ | $0.33 \pm 0.17$ | $0.22 \pm 0.15$ | $0.00 \pm 0.00$ | $0.11 \pm 0.11$ |
| Consensus | $0.89 \pm 0.11$ | $0.56 \pm 0.18$ | $0.22 \pm 0.15$ | $0.44 \pm 0.18$ | $0.11 \pm 0.11$ | $0.22 \pm 0.15$ |
| Leader Election | $0.67 \pm 0.17$ | $0.33 \pm 0.17$ | $0.44 \pm 0.18$ | $0.44 \pm 0.18$ | $0.44 \pm 0.18$ | $0.33 \pm 0.17$ |
| Matching | $0.11 \pm 0.11$ | $0.44 \pm 0.18$ | $0.22 \pm 0.15$ | $0.00 \pm 0.00$ | $0.00 \pm 0.00$ | $0.00 \pm 0.00$ |
| Vertex Cover | $0.33 \pm 0.33$ | $0.22 \pm 0.15$ | $0.00 \pm 0.00$ | $0.00 \pm 0.00$ | $0.00 \pm 0.00$ | $0.00 \pm 0.00$ |

Table 6: Byzantine fault tolerance performance for Gemini 2.0 Flash under adversarial conditions. Success rates reported as mean $\pm$ standard error, evaluated exclusively on honest agent responses.

| | 4 nodes | | 8 nodes | | 16 nodes | |
|---|---|---|---|---|---|---|
| Task | Aware | Not Aware | Aware | Not Aware | Aware | Not Aware |
| Coloring | $0.56 \pm 0.18$ | $0.78 \pm 0.15$ | $0.44 \pm 0.18$ | $0.67 \pm 0.17$ | $0.00 \pm 0.00$ | $0.22 \pm 0.15$ |
| Consensus | $0.67 \pm 0.17$ | $0.67 \pm 0.17$ | $0.33 \pm 0.17$ | $0.56 \pm 0.18$ | $0.00 \pm 0.00$ | $0.56 \pm 0.18$ |
| Leader Election | $0.56 \pm 0.18$ | $0.44 \pm 0.18$ | $0.00 \pm 0.00$ | $0.33 \pm 0.17$ | $0.00 \pm 0.00$ | $0.22 \pm 0.15$ |
| Matching | $0.67 \pm 0.17$ | $0.44 \pm 0.18$ | $0.22 \pm 0.15$ | $0.00 \pm 0.00$ | $0.00 \pm 0.00$ | $0.33 \pm 0.17$ |
| Vertex Cover | $0.56 \pm 0.18$ | $0.22 \pm 0.15$ | $0.11 \pm 0.11$ | $0.22 \pm 0.15$ | $0.00 \pm 0.00$ | $0.00 \pm 0.00$ |

```
[prompt for byzantine agents]

Your goal is to DISRUPT and SABOTAGE the other agents from
successfully completing their task.  You should try to:
send misleading or false information, contradict what other
agents are saying, propose incorrect solutions, create
confusion and disagreement, prevent consensus from being
reached, and make it harder for honest agents to coordinate.
```

Following the standard in Byzantine fault-tolerant systems, we assess task completion success exclusively based on the outputs of honest agents, treating Byzantine agent responses as extraneous to the coordination objective (**?**). We maintain the classical theoretical assumption $f < n/3$, where $f$ is the number of byzantine agents and $n$ the number of total agents (1 Byzantine agent out of 4, 2 out of 8, and 3 out of 16 total agents in our setup). We investigate two variations of the setup: (1) *Byzantine-aware*, where honest agents receive explicit notification of potential adversarial presence, and (2) *Byzantine-unaware*, where honest agents operate under standard AGENTSNET assumptions without knowledge of adversarial behavior.

### I.2 RESULTS AND ANALYSIS

The Byzantine fault tolerance evaluation reveals insights regarding the resilience characteristics of contemporary LLM-based coordination mechanisms. Performance metrics are presented for both Llama Maverick (Table 5) and Gemini 2.0 Flash (Table 6) across multiple network scales. The results show a systematic performance disparity between global coordination primitives and local combinatorial tasks under Byzantine adversarial pressure. Global coordination tasks (Consensus and Leader Election) exhibit better fault tolerance characteristics and maintain non-trivial success rates even under substantial adversarial presence. This phenomenon aligns with theoretical predictions from distributed computing literature: local optimization problems suffer disproportionately from Byzantine manipulation due to their dependency on immediate neighborhood integrity, while global coordination can potentially leverage distributed verification mechanisms and majority-based validation protocols.

Table 7: Prompt variation ablation results for Llama Maverick across all AGENTSNET tasks. Success rates reported as mean $\pm$ standard error for eleven prompt formulations, demonstrating task-dependent sensitivity to linguistic framing and validating the robustness of our standard prompt design.

| Prompt Variation | Coloring | Consensus | Leader Election | Matching | Vertex Cover | Average |
|---|---|---|---|---|---|---|
| Standard | $0.33 \pm 0.17$ | $0.78 \pm 0.15$ | $0.78 \pm 0.15$ | $0.22 \pm 0.15$ | $0.11 \pm 0.11$ | $0.44 \pm 0.07$ |
| Minimal | $0.22 \pm 0.15$ | $0.33 \pm 0.17$ | $0.56 \pm 0.18$ | $0.00 \pm 0.00$ | $0.67 \pm 0.33$ | $0.31 \pm 0.07$ |
| Step-by-step | $0.22 \pm 0.15$ | $0.78 \pm 0.15$ | $0.56 \pm 0.18$ | $0.33 \pm 0.17$ | $0.14 \pm 0.14$ | $0.42 \pm 0.08$ |
| Formal | $0.00 \pm 0.00$ | $0.33 \pm 0.17$ | $0.56 \pm 0.18$ | $0.00 \pm 0.00$ | $0.44 \pm 0.18$ | $0.27 \pm 0.07$ |
| Conversational | $0.11 \pm 0.11$ | $0.78 \pm 0.15$ | $0.67 \pm 0.17$ | $0.33 \pm 0.17$ | $0.50 \pm 0.50$ | $0.47 \pm 0.08$ |
| Imperative | $0.33 \pm 0.17$ | $0.33 \pm 0.17$ | $0.44 \pm 0.18$ | $0.11 \pm 0.11$ | $0.00 \pm 0.00$ | $0.24 \pm 0.07$ |
| Collaborative | $0.00 \pm 0.00$ | $0.33 \pm 0.17$ | $0.11 \pm 0.11$ | $0.00 \pm 0.00$ | $0.44 \pm 0.18$ | $0.18 \pm 0.06$ |
| Abstract | $0.22 \pm 0.15$ | $1.00 \pm 0.00$ | $0.00 \pm 0.00$ | $0.00 \pm 0.00$ | $0.22 \pm 0.15$ | $0.29 \pm 0.07$ |
| Real-World | $0.00 \pm 0.00$ | $1.00 \pm 0.00$ | $0.00 \pm 0.00$ | $0.00 \pm 0.00$ | $0.22 \pm 0.15$ | $0.24 \pm 0.07$ |
| Game-Theoretic | $0.00 \pm 0.00$ | $0.44 \pm 0.18$ | $0.00 \pm 0.00$ | $0.00 \pm 0.00$ | $0.22 \pm 0.15$ | $0.13 \pm 0.05$ |
| Resource-Constrained | $0.00 \pm 0.00$ | $0.78 \pm 0.15$ | $0.33 \pm 0.17$ | $0.00 \pm 0.00$ | $0.33 \pm 0.17$ | $0.29 \pm 0.07$ |

Contrary to intuition, informing honest agents about the presence of Byzantine adversaries does not produce systematic performance improvements under experimental conditions. In multiple task-model configurations, Byzantine-aware protocols exhibit a decrease in coordination effectiveness compared to uninformed baselines. This could suggest that current LLMs lack adversarial reasoning capabilities and may suffer from over-conservative coordination strategies when explicitly primed for adversarial scenarios, which leads to coordination failure through excessive suspicion rather than enhanced robustness.

These results establish that while LLM agents demonstrate reasonable coordination capabilities in benign environments, their robustness under adversarial conditions remains severely constrained. The absence of effective Byzantine fault tolerance mechanisms represents a critical vulnerability for real-world deployment of large-scale LLM-based multi-agent systems, particularly in security-sensitive applications where adversarial behavior is anticipated. This evaluation demonstrates the necessity of developing principled approaches to adversarial robustness in distributed LLM systems and highlights Byzantine fault tolerance as a future direction for scalable multi-agent AI architectures.

## J  PROMPT ABLATION

To investigate the sensitivity of AGENTSNET performance to prompt formulation and to ensure that our experimental design choices are well justified, we conduct a systematic ablation study examining the impact of different prompt styles on coordination effectiveness.

### J.1  EXPERIMENTAL DESIGN

We evaluated 11 distinct prompt variations on Gemini 2.0 Flash, selected to represent a diverse range of communication styles and task-framing approaches. The prompt variants are designed as follows:

- **Standard**: The original prompt formulation used throughout our main experiments

- **Minimal**: Reduced to essential information only (*"You are agent X. Communicate with neighbors. Output JSON."*)

- **Step-by-step**: Explicit procedural guidance (*"STEP 1: Analyze, STEP 2: Plan..."*)

- **Formal**: Academic and technical language (*"System specification", "algorithmic constraints"*)

- **Conversational**: Informal, friendly tone (*"Hey there! Think of it like a group project..."*)

- **Imperative**: Direct command structure (*"EXECUTE role. SEND messages. COMPLETE task."*)

- **Collaborative**: Emphasis on teamwork (*"Our collective success depends..."*)

- **Abstract**: Pure graph-theoretic framing (*"Node X in graph G, adjacency set..."*)

Table 8: Prompt variation ablation results for Gemini 2.0 Flash across all AGENTSNET tasks. Success rates reported as mean $\pm$ standard error, revealing model-specific sensitivity patterns to prompt formulation and demonstrating that our standard prompt provides robust baseline performance for fair model comparison.

| Prompt Variation | Coloring | Consensus | Leader Election | Matching | Vertex Cover | Average |
|---|---|---|---|---|---|---|
| Standard | $0.37 \pm 0.10$ | $0.89 \pm 0.06$ | $0.67 \pm 0.09$ | $0.33 \pm 0.09$ | $0.11 \pm 0.06$ | $0.47 \pm 0.04$ |
| Minimal | $0.33 \pm 0.09$ | $1.00 \pm 0.00$ | $0.44 \pm 0.10$ | $0.48 \pm 0.10$ | $0.11 \pm 0.06$ | $0.47 \pm 0.04$ |
| Step-by-step | $0.41 \pm 0.10$ | $0.96 \pm 0.04$ | $0.56 \pm 0.10$ | $0.56 \pm 0.10$ | $0.04 \pm 0.04$ | $0.50 \pm 0.04$ |
| Formal | $0.26 \pm 0.09$ | $0.93 \pm 0.05$ | $0.37 \pm 0.10$ | $0.11 \pm 0.06$ | $0.15 \pm 0.07$ | $0.36 \pm 0.04$ |
| Conversational | $0.37 \pm 0.10$ | $0.96 \pm 0.04$ | $0.44 \pm 0.10$ | $0.37 \pm 0.10$ | $0.04 \pm 0.04$ | $0.44 \pm 0.04$ |
| Imperative | $0.11 \pm 0.06$ | $0.89 \pm 0.06$ | $0.44 \pm 0.10$ | $0.11 \pm 0.06$ | $0.07 \pm 0.05$ | $0.33 \pm 0.04$ |
| Collaborative | $0.04 \pm 0.04$ | $0.48 \pm 0.10$ | $0.15 \pm 0.07$ | $0.04 \pm 0.04$ | $0.11 \pm 0.06$ | $0.16 \pm 0.03$ |
| Abstract | $0.15 \pm 0.07$ | $0.59 \pm 0.10$ | $0.11 \pm 0.06$ | $0.15 \pm 0.07$ | $0.00 \pm 0.00$ | $0.23 \pm 0.04$ |
| Real-World | $0.18 \pm 0.08$ | $0.00 \pm 0.00$ | $0.00 \pm 0.00$ | $0.00 \pm 0.00$ | $0.26 \pm 0.09$ | $0.09 \pm 0.03$ |
| Game-Theoretic | $0.22 \pm 0.08$ | $0.15 \pm 0.07$ | $0.22 \pm 0.08$ | $0.00 \pm 0.00$ | $0.07 \pm 0.05$ | $0.13 \pm 0.03$ |

- **Real-world**: Concrete application context (*"You are sensor X monitoring infrastructure..."*)

- **Game-theoretic**: Strategic competition framing (*"You are player X, maximize utility..."*)

- **Resource-constrained**: Efficiency-focused language (*"LIMITED bandwidth, minimize overhead..."*)

Each prompt variant maintains the core task specifications and communication protocol while varying the linguistic style, motivational framing, and level of procedural guidance provided to agents.

## J.2 RESULTS

Tables 7 and 8 present the performance results across all prompt variations for Llama Maverick and Gemini 2.0 Flash, respectively.

**Robustness of standard prompt design.** The standard prompt formulation used throughout our main experiments consistently ranks among the top performing configurations in both tested models. For Gemini 2.0 Flash, the standard variant achieves This performance validates our original design choices and demonstrates that our main experimental results are not artifacts of sub-optimal prompt engineering.

**Task-dependent vulnerability to prompt formulation.** Different coordination primitives exhibit varying robustness to prompt style changes. Consensus tasks demonstrate remarkable stability across prompt variants for both models, maintaining high success rates regardless of framing. Conversely, tasks such as Vertex Cover and Matching show substantial performance fluctuations, with success rates varying depending on prompt formulation. This pattern suggests that some tasks are more linguistically fragile than others.

**Counter-productive effects of specialized framing.** Several prompt variants designed to enhance coordination actually degrade performance across both models. The collaborative variant consistently underperforms, despite explicitly emphasizing teamwork. Similarly, game-theoretic framing yields poor results for both models. These findings indicate that overly specialized prompt engineering can interfere with emergent coordination strategies in LLM-based multi-agent systems.

These results demonstrate that, while prompt design significantly influences coordination performance, our standard formulation provides a robust and well-balanced baseline for fair model comparison. The systematic variation observed across prompt styles underscores the critical importance of standardized evaluation protocols in multi-agent benchmarking, as different linguistic formulations could systematically bias results in favor of particular model architectures. Our ablation study validates the methodological soundness of AGENTSNET, while revealing information on the linguistic factors that modulate coordination effectiveness in contemporary LLM-based distributed systems.

Table 9: Results of classical (randomized) algorithm baselines on AGENTSNET. Success rates reported as mean ± standard error.

| Task | 4 nodes | 8 nodes | 16 nodes |
|---|---|---|---|
| Coloring | 0.67 ± 0.17 | 0.67 ± 0.17 | 0.78 ± 0.15 |
| Consensus | 1.00 ± 0.00 | 1.00 ± 0.00 | 1.00 ± 0.00 |
| Leader Election | 1.00 ± 0.00 | 1.00 ± 0.00 | 1.00 ± 0.00 |
| Matching | 0.89 ± 0.11 | 0.89 ± 0.11 | 0.78 ± 0.15 |
| Vertex Cover | 1.00 ± 0.00 | 1.00 ± 0.00 | 1.00 ± 0.00 |

Table 10: Results of classical algorithm baselines on AGENTSNET up to 100 nodes. Success rates reported as mean ± standard error.

| Task | 20 nodes | 30 nodes | 40 nodes | 50 nodes | 60 nodes | 70 nodes | 80 nodes | 90 nodes | 100 nodes |
|---|---|---|---|---|---|---|---|---|---|
| Coloring | 0.89±0.11 | 0.78±0.15 | 0.78±0.15 | 0.67±0.17 | 1.00±0.00 | 1.00±0.00 | 0.56±0.17 | 0.89±0.11 | 0.56±0.17 |
| Consensus | 1.00±0.00 | 1.00±0.00 | 1.00±0.00 | 1.00±0.00 | 1.00±0.00 | 1.00±0.00 | 1.00±0.00 | 1.00±0.00 | 1.00±0.00 |
| Leader Election | 1.00±0.00 | 1.00±0.00 | 1.00±0.00 | 1.00±0.00 | 1.00±0.00 | 1.00±0.00 | 1.00±0.00 | 1.00±0.00 | 1.00±0.00 |
| Matching | 1.00±0.00 | 1.00±0.00 | 1.00±0.00 | 1.00±0.00 | 1.00±0.00 | 1.00±0.00 | 1.00±0.00 | 1.00±0.00 | 1.00±0.00 |
| Vertex Cover | 1.00±0.00 | 1.00±0.00 | 1.00±0.00 | 1.00±0.00 | 1.00±0.00 | 1.00±0.00 | 1.00±0.00 | 1.00±0.00 | 1.00±0.00 |

# K  CLASSICAL ALGORITHM BASELINES

To understand the performance gap between LLM-based systems and classical (randomized) algorithms given the selected round budgets, we present results on AGENTSNET with classical algorithmic baselines. Concretely, our algorithm implementations for Coloring, Vertex Cover, and Matching are all based on Luby's algorithm (Luby, 1985). Further, we solve Leader Election with a lexicographical sort. For Consensus, each agent initially samples a value at random. At each message-passing step, each agent receives the current value from its neighbors and updates its value according to the minimum value seen so far. These algorithms are implemented in the same python framework as AGENTSNET, meaning that the classical algorithms adhere to the same synchronous communication protocol and have the same round budget.

The results are presented in Table 9. We find that our implementations are consistently better than the results achieved by the agents tested in the paper, which means that there is significant room for future agents to improve performance. Further, when increasing the number of rounds, we find that the classical algorithms are able to perfectly solve AGENTSNET. As we apply the classical algorithms to the larger graph instances, we see the gap between LLM-based agents and classical algorithms increase; see Table 10. We find that the classical algorithms do not show a decrease in performance as the number of agents increases. In contrast, our experiment in Section 5.2 revealed LLM performance to sharply decline as the number of agents is increased.

Table 11: Token consumption statistics across models and network sizes. Values represent average tokens per instance over all AGENTSNET tasks.

| Model | 4 nodes | | | 8 nodes | | | 16 nodes | | |
|---|---|---|---|---|---|---|---|---|---|
| | Input | Output | Total | Input | Output | Total | Input | Output | Total |
| Claude 3.5 Haiku | 35.1K | 7.9K | 43.0K | 128.4K | 24.2K | 152.6K | 448.0K | 68.3K | 516.3K |
| Claude 3.7 Sonnet | 43.5K | 9.7K | 53.2K | 223.5K | 37.4K | 261.0K | 917.9K | 111.0K | 1029K |
| GPT-4.1 mini | 28.0K | 5.1K | 33.1K | 101.4K | 15.8K | 117.3K | 341.5K | 42.8K | 384.2K |
| Gemini 2.0 Flash | 25.9K | 4.1K | 30.0K | 95.9K | 13.8K | 109.7K | 333.0K | 39.2K | 372.1K |
| Gemini 2.0 Flash Thinking | 44.4K | 10.4K | 54.8K | 190.7K | 34.9K | 225.6K | 743.5K | 99.5K | 843.1K |
| Gemini 2.5 Flash | 44.8K | 14.2K | 59.0K | 206.7K | 54.8K | 261.5K | 723.4K | 133.4K | 856.8K |
| Gemini 2.5 Flash Thinking | 47.7K | 11.6K | 59.3K | 213.8K | 42.0K | 255.8K | 750.8K | 112.5K | 863.4K |
| Gemini 2.5 Pro | 58.6K | 17.9K | 76.5K | 266.3K | 62.2K | 328.5K | 936.6K | 157.6K | 1094K |
| Llama 4 Maverick | 32.0K | 8.0K | 40.0K | 107.9K | 22.0K | 129.9K | 356.8K | 57.2K | 414.0K |
| Llama 4 Scout | 33.3K | 7.2K | 40.6K | 111.4K | 19.6K | 131.1K | 370.4K | 51.4K | 421.8K |
| o4-mini | 26.6K | 20.7K | 47.3K | 104.5K | 64.3K | 168.7K | 367.8K | 177.0K | 544.8K |

## L  TOKEN CONSUMPTION ANALYSIS

Token efficiency represents a critical practical consideration for deploying large-scale multi-agent systems, as coordination complexity directly impacts inference costs. To quantify this relationship, we measure token usage statistics across all models and network configurations in AGENTSNET. Table 11 reports token consumption averaged per instance across all tasks, with usage statistics formatted as input tokens / output tokens / total tokens.

