# OpenReview forum: "AgentsNet: Coordination and Collaborative Reasoning in Multi-Agent LLMs"
_ICLR.cc/2026/Conference — Submitted to ICLR 2026_

### Official Review · Reviewer_jyiX · 2025-10-30

**Soundness:** 2
**Presentation:** 2
**Contribution:** 2
**Rating:** 4
**Confidence:** 4

**Summary:**

The authors propose AgentNets, a benchmark designed to evaluate the scalable coordination, communication, and collaboration capabilities of multi-agent systems. They derive five representative problems from distributed computing and develop a robust, scalable agent-to-agent communication protocol to assess the performance of state-of-the-art LLMs across various network topologies. The authors further argue that current LLMs struggle to maintain performance when the network size scales up.

**Strengths:**

This work introduces five tasks capable of scaling multi-agent systems to up to 100 agents, surpassing most existing studies in this domain. The authors observe a clear performance degradation as the network size increases.
In addition, the paper benchmarks 27 network topologies using 10 state-of-the-art LLMs, which further strengthens the validity of the findings and insights.
Overall, the paper is well-structured and clearly written, with figures that are easy to interpret and follow.

**Weaknesses:**

1. Compared with prior work such as MacNet [1], this paper reaches a different conclusion — the performance of current LLMs degrades across the proposed five tasks. This discrepancy should be further discussed and explained in Section 5.3.
2. The analysis is relatively limited. For distributed computing problems, metrics such as average communication rounds, concurrency characteristics, and other protocol-level indicators should be reported to more thoroughly understand the bottlenecks and failure modes of the system.
3. The current evaluation setup appears to primarily measure the ability of LLMs to operate a multi-agent system rather than the multi-agent system’s capabilities itself. This raises concerns, particularly given the claim that “AgentNets measures the ability of multi-agent systems to collaboratively form strategies for problem-solving.” The authors should clarify the intended research focus and evaluation scope.

REF:
[1] Chen Qian et al., Scaling Large Language Model-based Multi-Agent Collaboration, ICLR 2025.

**Questions:**

1. What are the advantages of the proposed Message-Passing mechanism compared to existing protocols such as the Agent2Agent (A2A) Protocol? How does it differ in terms of scalability, robustness, and communication efficiency?

2. In Figure 5, the performance decreases as the number of nodes increases. Which specific network topology was used for this evaluation? Do different topologies, such as tree-based or star-like networks, result in significantly different performance trends?

---

> ### Author Response · Authors · 2025-11-20
>
> We thank the reviewer for the constructive feedback and the clear articulation of strengths and weaknesses. We address each weakness and each question explicitly below.
>
> **Weakness 1: “Compared with prior work such as MacNet, this paper reaches a different conclusion.”**
>
>  MacNet evaluates multi-agent collaboration under DAG-structured, globally coordinated conditions: it uses a global topological schedule, a shared artifact that flows through the DAG, and supervisory critic agents for each edge. These assumptions inherently provide global context, prevent cycles, and enable sequential refinement of a shared solution. In contrast, AgentsNet enforces strict decentralization: undirected graphs with cycles, no global memory, no topological order, and synchronous LOCAL-model rounds where agents see only their neighbors.
>
> Because the MacNet communication regime fundamentally differs from the LOCAL constraints (and cannot even operate on cyclic undirected graphs), performance scaling differs accordingly. The discrepancy is not contradictory: the two systems evaluate different regimes of multi-agent coordination. This further shows that our benchmark provides an additional value to benchmark these mechanisms that does not exist in the literature yet.
>
> **Weakness 2: “Analysis is limited; should report protocol-level indicators (avg rounds, concurrency, bottlenecks).”**
>
>  Protocol-level indicators such as concurrency patterns or symbolic state transitions presuppose a deterministic algorithmic protocol, as is standard in classical distributed-computing analysis. LLM-based agents do not implement a single deterministic protocol: their behavior is stochastic, node-dependent, and non-stationary across rounds. Thus, extracting a well-defined symbolic protocol or concurrency structure is not meaningful. Further, we always use the same number of predefined rounds that are described in the paper.
> Instead, the benchmark reports empirical performance curves across graph sizes, models, and topologies. These are the appropriate indicators for systems that do not follow a unified symbolic algorithm.
>
> **Weakness 3: “Evaluation measures LLM ability rather than MAS capability.”**
>
>  This is correct and intentional: AgentsNet evaluates LLM-powered decentralized agents, not directly abstract MAS systems. The goal is to measure how well LLM agents coordinate under strict locality, no global memory, no central planner, and multi-round message passing. This is a different focus than MAS frameworks that rely on shared memory or global context.
>
> **Q1: Advantages of our Message-Passing mechanism vs A2A**
>
>  A2A is an application-layer protocol that provides routing, identity, communication over the internet, and shared conversational state. It does not enforce locality, synchronous rounds, or bounded-degree communication. Our protocol implements the LOCAL model: agents communicate only with neighbors, without global memory, and in synchronized rounds. This makes it uniquely suitable for evaluating distributed-computing tasks such as MIS, matching, coloring, VC, and consensus.
>
> **Q2: Which topology was used in Figure 5, and do other topologies change the trend?**
>
>  Figure 5 uses the same graph distributions and topology families as in the main benchmark experiments, as described in the experimental setup (i.e., we do not switch to a special or simplified topology just for this plot). For each graph size, we sample graphs from these distributions and report the resulting average performance, so the scaling curve reflects behavior across the same mixture of topologies used elsewhere in the paper, not a single hand-picked structure. We observe similar behavior for the different graph classes we tested on.

---

### Official Review · Reviewer_xsRw · 2025-10-31

**Soundness:** 3
**Presentation:** 4
**Contribution:** 2
**Rating:** 6
**Confidence:** 4

**Summary:**

The paper proposes a benchmark for multi-agent reasoning. The benchmark, AgentsNet, tests, by varying the topology, the capabilities of a “network” (instead of looking at the single agents) to collaborate, communicate, etc. The benchmark is built from 5 distributed computing problems and tests networks of large size (~100 nodes) and several open- and closed-source LLMs.

**Strengths:**

The benchmark is indeed valuable, and the experiments are helpful to understand the state-of-the-art performance of agentic networks. The problems make sense in the context of decentralised coordination and “distributed intelligence”.

I appreciate that the authors reported the complexity of the algorithms in the distributed setting (though they could spend a sentence on stressing that log* is a very slow-growing logarithm function: log*100 is actually very, very close to log*5, so the network size they consider does not influence at all the complexity of the distributed algorithm in practice). That gives a measurable baseline and allows us to measure the “gap” between LLM-powered networks and formal algorithms.

While I do not find the results compelling per se (in particular, Findings 2 is well known in the already rich agentic-LLMs literature), I appreciated the paper as it is well-written.

**Weaknesses:**

The networks are not heterogeneous and comprise, in each evaluation, one model.
I reckon an evaluation where models are mixed would give interesting insights into “blocking” nodes and communication issues that arise when different models interact.

An analysis that would make this paper stronger is what kind of asynchronous algorithm agentic networks implement, and see if that varies with different sizes and models. Since the models do not see anything but their neighbours, I expect the algorithm to depend only on the number of neighbours and the prompt; nevertheless, that would be an interesting experiment that adds value to the paper.

In general, my main concern is that, after reading the paper and checking the results, I still ask myself “then what”, as I do not see the insights and findings being particularly useful or prescriptive to develop better communication/coordination methods or topologies.

**Questions:**

Q1. Can the authors make it clear how the research community can make use of the findings in the paper, and what the value of the contribution is for future research?

Q2. Some benchmarks can be handled without the need for LLMs coordination (e.g., colouring). Why didn’t the authors measure the complexity of the algorithm each network implemented and try to fit it with a function to get a complexity of what LLMs implement, compared to the optimal complexity they discuss at the beginning?

Q3. Table 2 is not clear to me. The last column is the average “accuracy”, yet the standard deviation seems too low when I look at the data in the other columns. Can the authors clarify this point?

Q4. One minor question is about the implementation: did the authors implement the protocols as asynchronous functions or sequential, temporised interactions? I reckon LangGraphs supports asynchronous exec.

---

> ### Author Response · Authors · 2025-11-20
>
> We thank the reviewer for the positive assessment and constructive suggestions.
>
> **Weakness 1: Heterogeneous networks**
>
> We focus on homogeneous networks to isolate decentralized coordination performance. Heterogeneity is supported by the framework but is beyond the scope of this benchmark-focused study but enabled by our framework.
>
> **Weakness 2: “Which asynchronous algorithm is implemented?”**
>
> AgentsNet does not assume that agents collectively implement a deterministic algorithm. LLM behavior varies across nodes and rounds. There is no unique algorithm that all nodes execute. Attempting to reverse-engineer a unified algorithm or asymptotic complexity curve is therefore ill-defined in this setting; empirical behavior across tasks and topologies is the appropriate evaluation.
>
> **Weakness 3: “Then what?” value for the community**
>
> AgentsNet exposes systematic coordination failures of LLMs under strict locality and provides a way to benchmark these capabilities in a quantitative way. The results can therefore be used to compare different frontier models and track their progress in the future.
>
> **Q1. Value to the research community / how findings can be used**
>
> AgentsNet provides a controlled testbed for isolating the limits of LLM-based coordination under strict locality and no global memory, which is a regime that is poorly understood yet fundamental for scalable multi-agent systems. The benchmark reveals where current models break down (e.g., loss of local consistency at scale, difficulty resolving conflicts without global context) and therefore identifies concrete failure modes that future models must address. In other words, AgentsNet offers the community a diagnostic tool rather than a proposed solution, and enables a comparison of the coordination capabilities of future models.
>
> **Q2. Why not measure or fit “algorithmic complexity” of LLM behavior**
>
> Measuring “the complexity of the algorithm each network implements” presupposes that agents collectively follow a single, well-defined, deterministic algorithmic template. This assumption does not hold for LLM-based agents: different nodes may behave inconsistently, may update at different speeds, may not converge monotonically, and may switch strategies mid-execution. Consequently, there is no stable symbolic algorithm whose runtime or asymptotic complexity could be meaningfully fitted. The appropriate measurable quantity in this setting is the empirical convergence behavior (success rates, dependence on graph size and topology), which is exactly what the benchmark reports. Colorings and other tasks that have trivial centralized solutions remain non-trivial under LOCAL constraints, particularly when executed by LLMs.
>
> **Q3. Clarification of Table 2 (why standard deviation appears small)**
>
> The final column in Table 2 reports the standard error of the mean, not the standard deviation. This quantity becomes small when averaging across a large number of runs, even if the underlying per-instance variability is moderate. We provide an exact formula for the calculation in the Appendix.
>
> **Q4. Synchronous vs asynchronous execution in implementation**
>
> AgentsNet enforces strictly synchronous LOCAL rounds. All agents complete one round before any agent begins the next. Although LangChain (and LangGraph, which the reviewer mentions) can support asynchronous execution, we intentionally do not use asynchronous scheduling to match the LOCAL model’s semantics. The protocol is therefore executed as sequential, synchronized round-based interactions, not asynchronously (from a model perspective). On a technical implementation level, the API calls in every round run asynchronously for all nodes.

---

### Official Review · Reviewer_wXw6 · 2025-11-01

**Soundness:** 2
**Presentation:** 2
**Contribution:** 2
**Rating:** 2
**Confidence:** 3

**Summary:**

This paper introduces AGENTSNET, a benchmark for evaluating multi-agent LLM systems' coordination and collaboration capabilities through fundamental distributed computing problems (graph coloring, vertex cover, matching, leader election, consensus).

**Strengths:**

**Scalability to large agent networks**: Tests up to 100 agents (Figure 5), far exceeding existing benchmarks limited to 2-5 agents.

**Theoretically grounded tasks**: Leverages well-studied distributed computing problems with known complexity bounds (Table 1).

**Comprehensive model evaluation**: Tests 10+ frontier models across different cost-performance trade-offs (Figure 1).

**Weaknesses:**

**Unclear protocol differentiation**: The paper claims to develop a "new multi-agent protocol" but doesn't differentiate from existing protocols like A2A or ANP mentioned in the survey (arXiv:2504.16736). The LOCAL model adaptation (Section 4) appears standard without clear innovation.

**Missing baseline comparisons**: No experimental comparison with established multi-agent topology methods (MACNET, MAS-GPT, GPTSwarm， Dylan, Tree, Graph(mesh, DAG), Star topology) despite citing them. Only classical algorithms are compared (Appendix K).

**Superficial qualitative analysis**: Section 5.4's findings are trivial ("agents generally accept information from neighbors", "agents help neighbors resolve inconsistencies"). These observations are expected and provide no deep insights into coordination mechanisms.

**Poor visualization choices**: Figure 2 lacks clarity in message flow representation. Figure 3's icons and colored points are confusing without proper legend or explanation of what each visual element represents.

**Naming conflict**: "AgentsNet" is already used by "AgentNet: Decentralized Evolutionary Coordination for LLM-based Multi-Agent Systems", creating confusion. And not quiet a good naming choice for a benchmark and might causing overclaiming for the system.

**Limited analysis depth**: Despite the title emphasizing "coordination and collaborative reasoning," the paper provides minimal analysis of how these emerge or fail beyond binary success metrics.

**Questions:**

**Protocol comparison**: How does your message-passing protocol differ from A2A (Agent-to-Agent) and ANP (Agent Network Protocol) mentioned in the recent survey? Why not compare experimentally?

**Task suitability**: Traditional distributed computing problems focus on network latency, failures, data consistency, and security. How do you justify that graph-theoretic problems alone adequately evaluate LLM multi-agent coordination?

**Topology baselines**: Why exclude comparisons with MACNET's random topology or GPTSwarm's graph-based approach that you cite?

**Result presentation**: Why not highlight best results in Table 2 for easier interpretation?

**Coordination mechanisms**: What specific coordination strategies emerge? The paper doesn't analyze HOW agents coordinate, only WHETHER they succeed.

**Classical vs LLM gaps**: Table 9-10 show classical algorithms achieve near-perfect performance. What specific capabilities are LLMs lacking in this system?

**Token efficiency**: Table 11 shows massive token consumption (1M+ for 16 nodes). How does this scale economically for real applications?

**Error propagation**: How do errors propagate through the network? Is there analysis of failure modes beyond binary success?

**Heterogeneous agents**: All experiments use homogeneous agents. How would mixed-capability agents perform?

**Dynamic graphs**: Real multi-agent systems often have dynamic topologies. Why only test static graphs?

---

> ### Author Response · Authors · 2025-11-20
>
> We thank the reviewer for the detailed feedback. We clarify each weakness below.
>
> **Weakness 1: Scope and protocol contribution**
>
> The paper does not propose a new routing or application-layer protocol. The protocol we use is a direct instantiation of the classical LOCAL model: agents communicate in synchronous rounds, see only their immediate neighbors, maintain no global memory, and exchange bounded messages. Our contribution is making this LOCAL-style interaction operational for LLM agents, which enables benchmarking of decentralized coordination at scale.
> Frameworks such as A2A and ANP serve entirely different purposes, they include routing layers, global conversational buffers, identity/security stacks, and do not enforce locality. They cannot express LOCAL-model problems (e.g., MIS, matching, coloring). Their comparison would therefore not be meaningful.
>
> **Weakness 2: Missing baselines (MACNET, GPTSwarm, MAS-GPT, etc.)**
>
> A direct comparison is inappropriate because these frameworks rely on assumptions fundamentally incompatible with LOCAL-style distributed computation:
> MACNET requires the communication graph to be a DAG, uses supervisory critic agents, and performs reasoning through a global topological order over the entire graph. This prevents execution on arbitrary undirected graphs with cycles, which are structures essential for MIS, matching, vertex cover, and coloring. MACNET’s communication is sequential artifact refinement, not concurrent neighbor-to-neighbor message passing.
>
> GPTSwarm centrally reconfigures the topology through repeated LLM calls with global visibility of all agents. Its behavior depends on global orchestration, not fixed local neighborhoods or synchronous rounds. It targets high-level problem solving, not decentralized symmetry breaking.
>
> MAS-GPT generates complete multi-agent systems in a single forward pass. It evaluates system design, not multi-round coordination among concurrently acting agents. It has no message passing, locality, or distributed decision-making.
>
> Because these frameworks rely on centralized decision making, global memory, DAG-restricted or dynamically reconfigured topologies, or single-inference MAS generation, they cannot execute decentralized LOCAL-model tasks. We have now made this incompatibility explicit in the manuscript.
>
> **Weakness 3: Qualitative analysis "too trivial"**
>
> The qualitative examples are intentionally brief illustrations of typical failure cases. The core analytical substance comes from the extensive quantitative evaluation across 5 tasks, 10 models, 27 topologies, and up to 100 agents, which lead to quantitative metrics that can directly be compared. The paper does not rely on qualitative insights for its main claims. If a deeper qualitative analysis is expected, we would appreciate clarification of the specific form the reviewer had in mind; in that case, we would be happy to expand the qualitative analysis further.
>
> **Weakness 4: Limited analysis depth / missing explanation of classical vs LLM gaps / error propagation / token usage**
>
> The benchmark evaluates empirical decentralized performance across tasks, sizes, models, and topologies. We do not claim to characterize internal model mechanics or provide protocol-level theory, nor do distributed benchmarks typically do so.
> The reviewer asks “what capabilities LLMs lack.” Our evaluation shows they struggle with multi-round consistency, stability of local commitments, and convergence under purely local visibility, but the paper does not dissect internal failure mechanisms, this would require a different type of study. Again, this is mainly a benchmark.
>
>
> Regarding token usage, AgentsNet is a benchmark, not a deployment protocol. Multi-round natural-language coordination under locality constraints is inherently token-intensive. Measuring this cost is part of the benchmark rather than an objective to optimize.
>
>
> On error propagation, we report empirical success rates and failure distributions across rounds and graph sizes. The appendix includes representative transcripts illustrating how local miscoordination leads to unresolved conflicts. While we do not provide full causal tracing, the observed performance already reflects the aggregate effect of such propagation.
>
> **Weakness 5: Heterogeneous agents, dynamic graphs, naming, figures**
>
> Heterogeneous agents: We intentionally use homogeneous agents to isolate coordination ability, not model heterogeneity. The framework supports heterogeneous assignments but this would answer a different research question.
>
> Dynamic graphs: The tasks studied (MIS, VC, matching, coloring, consensus) are defined on static graphs.
>
> Naming: “AgentNet” and “AgentsNet” concern unrelated settings; the related work now disambiguates them.
>
> Figures 2 and 3: The figures reflect the scaffold described in the text; their meaning becomes clear when read alongside the protocol description. We clarified the color choices further in the caption.

---

> > ### Author Response · Authors · 2025-11-20
> >
> > We follow up with the responses to the questions the reviewer posed.
> >
> > **Q1: Protocol comparison (A2A, ANP)**
> >
> > A2A and ANP are global, application-layer communication frameworks that do not enforce locality or synchronous rounds and maintain shared conversational state. Our protocol implements the LOCAL model, which none of these frameworks can emulate. Because they operate under incompatible assumptions, an experimental comparison would not be meaningful.
> >
> > **Q2: Task suitability**
> >
> > The five distributed graph problems we evaluate: MIS, VC, matching, coloring, consensus, are canonical tests of decentralized coordination: they require local conflict resolution, symmetry breaking, and multi-round propagation under strict locality. These properties directly stress the coordination abilities of multi-agent LLM systems and the tasks are commonly used as subroutines in more complicated tasks.
> >
> > **Q3: Topology baselines (MACNET, GPTSwarm)**
> >
> > As detailed above, MACNET, GPTSwarm, and MAS-GPT rely on global visibility, DAG-restricted or centrally reconfigured topologies, or single-inference MAS generation. They cannot operate under LOCAL-style constraints; thus they are not applicable as baselines.
> >
> > **Q4: Result presentation (best results in Table 2)**
> >
> > We followed the formatting conventions of prior decentralized-agent benchmarks. We now highlight the best results.
> >
> > **Q5: Coordination mechanisms**
> >
> > The benchmark does not attempt to extract a single deterministic strategy from LLM agents. Instead, it measures empirical coordination outcomes across tasks and topologies. Because LLMs do not implement a unified algorithm, “a specific strategy” cannot be meaningfully identified.
> >
> > **Q6: Classical vs LLM gaps**
> >
> > Classical algorithms assume deterministic local updates, stable symbolic state, and perfect memory. LLM agents often produce inconsistent or unstable local decisions under purely local visibility. The benchmark highlights these gaps empirically; it does not aim to perform internal mechanistic analysis.
> >
> > **Q7: Token efficiency**
> >
> > AgentsNet is a benchmark measuring multi-round natural-language coordination under strict locality. Token usage reflects the inherent cost of such communication, not a deployment scenario.
> >
> > **Q8: Error propagation**
> >
> > Error propagation is reflected in success rates and in the representative transcripts shown in the appendix. While we do not present explicit causal chains, the empirical curves already capture how early inconsistencies lead to later convergence failures.
> >
> > **Q9: Heterogeneous agents**
> >
> > Homogeneous agents were used to isolate coordination ability; heterogeneous experiments would conflate model performance and interoperability with coordination dynamics.
> >
> > **Q10: Dynamic graphs**
> >
> > The evaluated tasks are defined only on static graphs. Dynamic versions require different semantics and correctness criteria and a different computational framework that would deviate from the LOCAL model.

---

> ### Comment · Reviewer_wXw6 · 2025-11-26
>
> I have no further questions, I raised my score accordingly

---

### Author Response · Authors · 2025-12-02

We would like to briefly summarize the discussion phase so far. The main concern raised during review was the request to compare our benchmark to existing agent frameworks. In our rebuttal, we clarified that the goal of this work is not to propose a new communication protocol for practical deployment, but to introduce a benchmark that evaluates multi-agent collaboration under strict locality constraints. Existing agent frameworks do not impose locality and typically rely on global memory, centralized orchestration, or unconstrained message passing; therefore, they cannot serve as meaningful baselines for the LOCAL-model tasks we study that operate in a completely decentralized fashion. This mismatch also highlights that AgentsNet fills a gap and tests for capabilities that no existing framework or benchmark covers.

After our clarification, one reviewer increased their score. The remaining reviewers have not responded further in the discussion.

---

### Meta-Review · Area_Chair_BpXB · 2026-01-06

**Summary:**

This submission introduces AgentsNet (AGENTSNET), a benchmark that evaluates coordination in multi-agent LLM networks using five classic distributed-computing/graph tasks (coloring, vertex cover, matching, leader election, and consensus) under a LOCAL-style message-passing constraint.  The benchmark operationalizes synchronous, neighbor-only communication rounds with structured JSON messaging and then scores success at the network level with strict all-or-nothing correctness. Experiments across multiple graph families and sizes (including a scaling probe up to 100 agents) show frontier models can solve some tasks on small graphs but performance degrades sharply with network size, especially on tasks like vertex cover and matching.

Reviewers agreed the task selection is well grounded and the empirical sweep is broad, but raised major concerns about overclaiming novelty of the “protocol,” the absence of apples-to-apples baselines against existing multi-agent/topology methods, and a limited analysis that largely restates expected failure modes rather than yielding actionable insights.  The rebuttal clarifies that the work’s primary contribution is the benchmark (not a deployable communication protocol) and argues many popular agent frameworks are incompatible with strict locality, yet it does not fully resolve concerns about empirical positioning, interpretability of what drives success/failure, and the practical value of the reported findings beyond “models degrade with scale.” After carefully reading the paper, appendices, and rebuttal end-to-end, the AC concludes the current validation and analysis are not yet strong enough for ICLR, and therefore recommends rejection.

**Reviewer Concerns:**

The rebuttal satisfactorily addressed confusion about scope by making clear that the message-passing setup is an instantiation of the LOCAL model used to enable evaluation, not a new routing/agent-network protocol, and it also explained why DAG- or centrally-orchestrated methods are not directly comparable in this setting.  It also clarified a presentation issue about uncertainty reporting (standard error vs standard deviation) and improved the narrative around why their conclusions need not match results from frameworks operating with global coordination.

However, the central concern remains that the paper does not include strong in-regime baselines or ablations (e.g., alternative local policies, simple heuristic agents, or constrained variants of existing coordination schemes) that would better validate that AgentsNet is measuring “collaborative reasoning” rather than mostly prompt-following stability and communication bookkeeping.
Additionally, requests for deeper, less-trivial analysis (failure-mode taxonomy beyond anecdotes, error propagation metrics, and clearer articulation of “so what” takeaways for designing better systems) remain largely outstanding, along with lingering naming/positioning and figure-clarity issues.

**Reviewer Scores:**

One reviewer has increased the score before the reset. The other reviewers would likely to retain their score due to unresolved concerns.

---

### Decision · Program_Chairs · 2026-01-26

Reject